# Discrete mechanical model of lamellipodial actin network implements molecular clutch mechanism and generates arcs and microspikes

**David M. Rutkowski**, **Dimitrios Vavylonis** *

Department of Physics, Lehigh University, Bethlehem, Pennsylvania

* vavylonis@lehigh.edu

**Data Availability Statement:** All relevant data are within the manuscript and its Supporting information files. The C++ code of the simulation is

## Abstract

Mechanical forces, actin filament turnover, and adhesion to the extracellular environment regulate lamellipodial protrusions. Computational and mathematical models at the continuum level have been used to investigate the molecular clutch mechanism, calculating the stress profile through the lamellipodium and around focal adhesions. However, the forces and deformations of individual actin filaments have not been considered while interactions between actin networks and actin bundles is not easily accounted with such methods. We develop a filament-level model of a lamellipodial actin network undergoing retrograde flow using 3D Brownian dynamics. Retrograde flow is promoted in simulations by pushing forces from the leading edge (due to actin polymerization), pulling forces (due to molecular motors), and opposed by viscous drag in cytoplasm and focal adhesions. Simulated networks have densities similar to measurements in prior electron micrographs. Connectivity between individual actin segments is maintained by permanent and dynamic crosslinkers. Remodeling of the network occurs via the addition of single actin filaments near the leading edge and via filament bond severing. We investigated how several parameters affect the stress distribution, network deformation and retrograde flow speed. The model captures the decrease in retrograde flow upon increase of focal adhesion strength. The stress profile changes from compression to extension across the leading edge, with regions of filament bending around focal adhesions. The model reproduces the observed reduction in retrograde flow speed upon exposure to cytochalasin D, which halts actin polymerization. Changes in crosslinker concentration and dynamics, as well as in the orientation pattern of newly added filaments demonstrate the model's ability to generate bundles of filaments perpendicular (actin arcs) or parallel (microspikes) to the protruding direction.

## Author summary

Cells adhered to flat surfaces can undergo directed motion by extending a sheet-like network of polymerizing actin filaments near the front of the cell known as the

available at https://github.com/davidmrutkowski/
LamellipodiumBrownian.

**Funding:** This work was supported by National
Institutes of Health Grant R01GM114201 and
R35GM136372 to DV. Use of the high-
performance computing capabilities of the Extreme
Science and Engineering Discovery Environment
(XSEDE), which is supported by the National
Science Foundation, project no. TG-MCB180021 is
also gratefully acknowledged. The funders had no
role in study design, data collection and analysis,
decision to publish, or preparation of the
manuscript.

**Competing interests:** The authors have declared
that no competing interests exist.

lamellipodium. The lamellipodium links to the environment outside the cell via regions
known as focal adhesions, which help convert the polymerization force into a protrusive
force that drives the cell membrane forward. Though forces on actin filaments have been
found to enhance filament severing, possibly affecting the structure of the lamellipodium,
the force balance within the lamellipodium at the filament level remains largely unex-
plored. We have determined the forces within a steady-state model of the lamellipodium
actin network in the presence of a focal adhesion simulated using 3D Brownian dynamics.
We recapitulate the main idea of the clutch model, which states that increased resistance
due to focal adhesions results in reduced retrograde flow and enhanced protrusion force.
Our simulations show that the lamellipodial network close to the leading edge is under
compression while further away the network is under extension possibly reflecting the
change seen in network topology. Finally, we find that by altering the angle of filament
introduction near the leading edge filament bundles similar to those seen in experiment
can form.

## Introduction

Cells adhered to 2D surfaces can undergo crawling motion by extending a thin, sheet-like net-
work of polymerizing actin near the leading edge of the cell known as the lamellipodium [1–
4]. The actin network within the lamellipodium is a highly branched network of filaments
which pushes against the plasma membrane at the leading edge as each filament polymerizes,
in order to advance the cell. Branches in the network form due to the Arp 2/3 complex, which,
after activation close to the leading edge, binds to the side of an existing an actin filament and
nucleates the growth of a new actin filament at an angle of $\sim 70°$ with respect to the pointed-
to-barbed end axis. At the back of the lamellipodium, the actin network converts back to
monomeric actin due to a combination of cofilin-mediated filament severing and depolymeri-
zation. Thus, lamellipodia exist in state of constant turnover, driven by the energy of ATP
hydrolysis bound to actin subunits that age as they hydrolyze bound ATP to ADP + $P_i$ and
release $P_i$ [2–5].

Crawling cells control motility by regulating the fraction of actin polymerization that is
converted into membrane protrusion at the leading edge versus retraction of the actin network
towards the interior of the cell (known as retrograde flow). The balance between protrusion
and retrograde flow is hypothesized to be controlled by a clutch mechanism: retrograde flow is
impeded by the presence of focal adhesions, connecting actin filaments in the lamellipodium
and the lamella (the actin cytoskeleton region behind the lamellipodium and closer to the cell
interior) to the external substrate via integrins and focal adhesion-associated proteins such as
talin and vinculin [6–8]. When the clutch is "engaged" (i.e. when the lamellipodial network is
strongly attached to the focal adhesion) the frictional force on the actin network increases lead-
ing to a reduction in retrograde flow speed and an increase in the force on the leading edge by
the lamellipodial network, helping to increase cell propulsion speed.

In addition to the forces of actin polymerization at the leading edge and forces at focal
adhesions, forces by molecular motors also aid retrograde flow. In lamellipodia of neuronal
growth cones, retraction forces are generated by myosin II motors in the central region of the
growth cone [9]. However, in many other cell types, forces are generated by retrograde flow
motors different than myosin II although the precise identity of these motors is unknown [10].

The actin network in lamellipodia is not uniform but has spatial and temporal variations in
morphology, the regulation of which occurs at the filament level in tandem with several actin

binding proteins. Close to the leading edge, the density of the Arp2/3 complex per filament is high, resulting in short branches that are capable of supporting force, a process that is likely reinforced by branching being biased to occur at curved filaments [11]. Further back towards the middle of the lamellipodium, electron microscopy of keratocyte cells showed that the network morphology changes to longer filaments with lower branch density [12]. This change in morphology is likely due in part to debranching of filaments which can occur due to hydrolysis and is enhanced by force on the branch [13]. Crosslinkers, including $\alpha$-actinin and filamin [14], are present throughout the lamellipodium and bind to and connect actin filaments. $\alpha$-actinin, filamin, and plastin are thought to increase the stiffness of the lamellipodium, especially for regions with low branch density, in order to strengthen the lamellipodium into a network capable of sustaining and exerting force [14–17]. Another crosslinker, fascin, can tightly bundle parallel actin into higher density microspikes / filopodia within the lamellipodia. These bundled structures can push against the leading edge to form finger-like protrusions [18, 19]. Bending of filaments in the lamellipodium may occur near obstacles to retrograde flow such as near focal adhesion regions [10]. Destruction of the lamellipodial network can be enhanced by severing of these bent filaments which occurs at higher frequency for filaments partially decorated with the actin binding protein cofilin [20].

The filament-level processes described above are demanding of quantitative models that can account for the feedback between biochemistry and filament mechanics. Such models can also be used to interpret the results of mechanical experiments such as those using traction force microscopy and measurements of cytoskeletal flow, which have so far been analyzed based on continuum approximations [21–26]. While prior models have investigated the structure and forces within the lamellipodium, many have simplified assumptions and do not include discrete filament force analysis over the entire lamellipodial network. Several models and simulations have investigated the mechanics of branched, bundled, and crosslinked networks [27–30], however these models were not applied to dynamic lamellipodia and largely consider static networks without filament addition. Some models of dynamic lamellipodia with network growth have used continuum approximations for the filamentous network [9, 31–35], or considered filaments attached to a continuous gel [36, 37]. Models with individual filaments provide the most detail, but several of these prior studies do not constitute a complete lamellipodium model since there is no interaction of retrograde flow with adhesions and there is no pulling forces due to motors [38–40]. Schreiber et al. [41] simulate a lamellipodium at the filament level with a model that included both the leading edge and a focal adhesion region; however, the lamellipodium network in this model lacked crosslinkers and consisted of short individual branches that did not form a continuous network so force propagation through the network was not investigated. Models focused on focal adhesions have also been considered in several studies focusing on understanding the interactions between adhesions and the lamellipodial actin network, however the filamentous network in these models was not treated explicitly [25, 42–44].

Here we model a 2 $\mu m$ wide strip of the lamellipodium actin network at the filament level, incorporating both pushing forces from the leading edge and pulling forces due to motor proteins, in the presence of a focal adhesion. Rather than focusing on the details of the specific sub-process of filament addition/crosslinking/destruction and binding to individual adhesion molecules, as has been investigated in several prior works, we instead create a steady state of the whole lamellipodium at the filament level, which can later be refined by improving local mechanisms. As a reference experimental system, we use XTC cells studied by Yamashiro et al. [10] who measured actin flow patterns in stationary lamellipodia, around both mature and nascent focal adhesions. We use this cell system because their lamellipodia are relatively wide (extending $\sim 5 \mu m$ into the cell) and have a nearly constant value of retrograde flow speed on

the order of 50 *nm/s* that was accurately measured by single molecule imaging, without complicating ruffling or contractile instabilities frequently observed in other cell types or different conditions. (XTC cells can also exhibit protrusion and retraction cycles of the leading edge at constant retrograde flow [45], however here we consider conditions where the leading edge is stationary.) With this model we investigate the effect of the focal adhesion strength on retrograde flow speed and internal forces where we tune the balance of pushing and pulling forces to approximately match the retrograde flow speed of XTC cells as a function of focal adhesion strength. We find that there is a region of higher than average filament bending near the focal adhesion region. Since we incorporate crosslinkers into our simulations we are able to explore the mechanisms of bundle formation in lamellipodia including microspikes/filopodia and stress fiber arcs.

## Results

We simulated an actin filament network representing the lamellipodium of a stationary cell with broad lamellipodia undergoing steady retrograde flow. We used XTC cells from the experiments of Yamashiro et al. [10] as a reference case to adjust the model's parameters. The model uses 3D Brownian dynamics simulations to evolve actin filaments under the forces of polymerization, motor pulling and interaction with focal adhesions (Fig 1A, and Modeling methods for details). We assume the leading edge is fixed at $y = 0$, a lamellipodium height of 0.2 $\mu m$, and apply periodic boundary conditions along the lamellipodium every 2 $\mu m$. Filaments are represented as beads connected by spring and bending forces (Fig 1B) and interact with each other via excluded volume interactions (Fig 1D).

Our aim is to generate a coarse filament-level representation that includes the main force contributions over the size of the lamellipodium, without at this point elaborating on scales on the order of the smallest element in the simulation ($\sim 100$ *nm*), with further improvements left for future work. In particular, we do not consider structural changes in the lamellipodium from a dendritic brushwork of short filaments close to the leading edge, to longer filaments in the back region [2, 12] and do not account for the distributed turnover of actin monomers occurring throughout the lamellipodium as revealed by single molecule microscopy [10, 46]. Recent work suggested that short actin lifetimes may be linked to actin network structural changes through severing and annealing near barbed ends [47]. Such processes occurring near filament ends may maintain a primarily elastic response of the actin network in the lamellipodium as assumed here. Our model thus differs from the purely viscous actin network model of [25].

We implement polymerization at the leading edge by adding individual filaments as 1 $\mu m$-long segments (a typical filament length in lamellipodia [12, 15, 48]), with their pointed ends at $y = 0.5$ $\mu m$ and orientations along the *xy* plane between -70˚ and 70˚, a distribution that approximately corresponds to the experimental distribution in fibroblast cells, however without peaks at -35˚ and 35˚ [15, 49]. In this way we leave out most of the complexity of force generation by polymerization close to the leading edge: the effect of the plasma membrane on the network is only included as a pushing force, without accounting for the force-elongation properties of membrane complexes that transfer actin or profilin-actin to the barbed ends of filaments depending on filament length and orientation [36, 50–52]. The force generation by polymerization was accounted for by a constant pushing force of 1.5 *pN* per filament applied to all filaments close to the leading edge. Because of the simplified assumptions involved with this force, the leading edge region is not sharply defined at $y = 0$, with boundary effects extending up to a maximum of 0.5 $\mu m$ into the cell (Fig 1A).

Permanent cross-links, such as those generated by the Arp2/3 complex, are added to newly inserted filaments, while short-lived dynamic cross-links are added between beads of

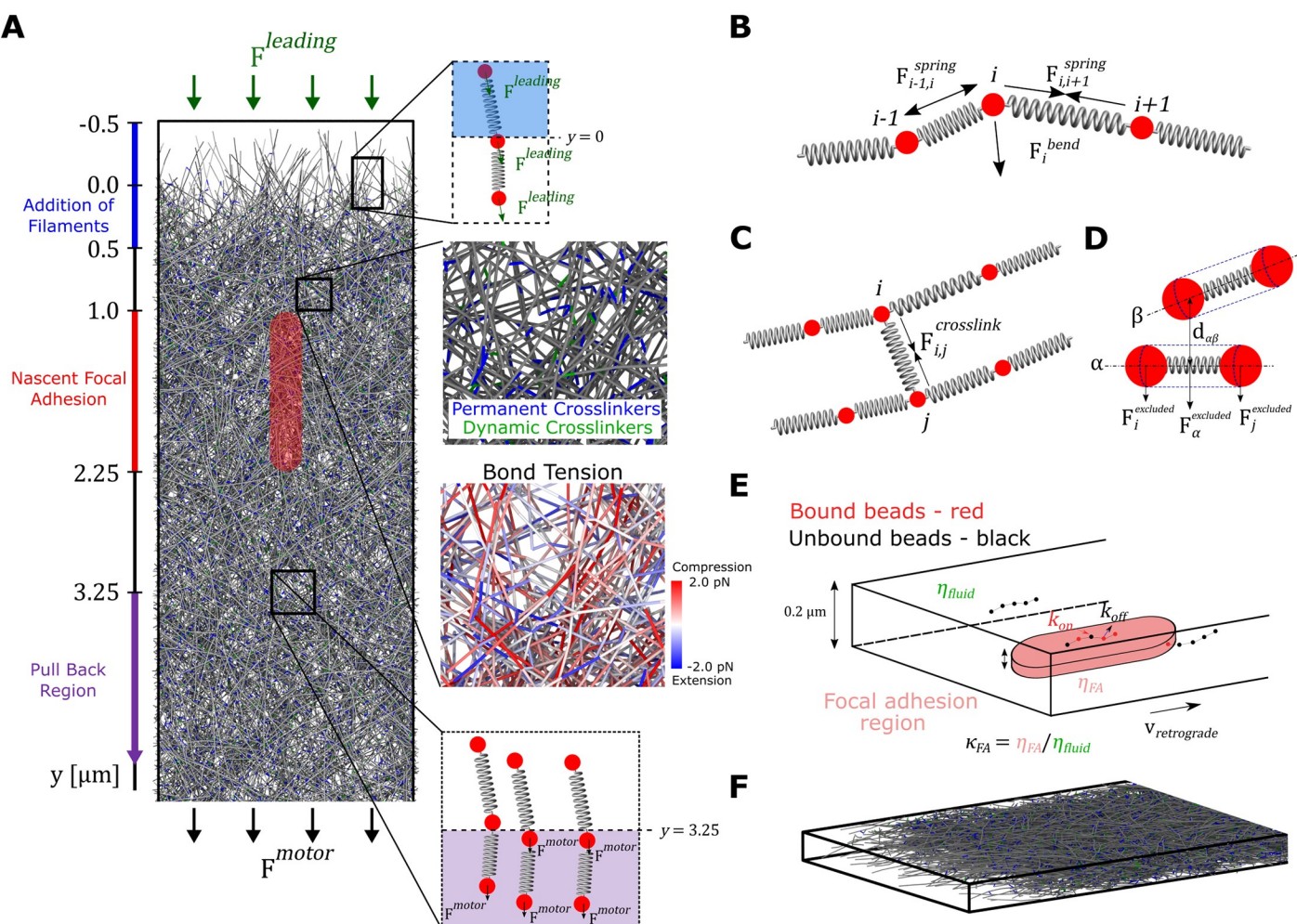

**Fig 1. Brownian dynamics simulations of a filament-level model of a lamellipodial actin network undergoing retrograde flow.** (A) The lamellipodial actin network is generated by the addition of 1 $\mu m$ long actin filaments (modeled as a linear chain of discrete beads connected via springs) close to the leading edge. The effect of polymerization against the leading edge is modeled as a force that pushes downwards on filaments close to the leading edge. The effect of molecular motors is modeled as a force that pulls actin beads downward. The molecular motor force either acts on all actin beads (uniform pulling) or only on actin beads with $y$ position larger than 3.25 $\mu m$ (back pulling). The red region indicates the location of the nascent focal adhesion within which filament beads experience a higher viscosity. As they age, filament segments are removed by bond severing near the bottom of the network. Zoomed in image displays the actin network in greater detail where permanent crosslinkers (representing Arp2/3 complex) are colored in blue and dynamic crosslinkers (representing $\alpha$-actinin, filamin or plastin) are colored in green. Another zoomed in image displays actin segments colored based on their instantaneous bond tension. (B) A spring force acts between neighboring pairs of beads in each actin filament to maintain an actin segment length of 0.1 $\mu m$. A bending force acts on triplets of neighboring beads corresponding to persistence length of 17 $\mu m$. (C) Dynamic and permanent crosslinks are represented as spring connection between filament beads. (D) Each actin filament segment experiences excluded volume interactions with nearby segments as indicated by the dashed blue line surrounding each spring. The excluded volume interaction between actin filament segments is calculated based on whether the minimum distance between two neighboring filament segments, $d_{\alpha\beta}$, is below a cutoff distance, $d^{excluded}$. The excluded volume force is distributed to the two beads at the end of the filament segment. (E) Beads within the focal adhesion region bind and unbind to the focal adhesion via a discrete-time Gillespie algorithm with defined on and off rates. When bound to the focal adhesion region beads experience a higher viscosity than the surrounding fluid. (F) Side view of model lamellipodium network showing 3D aspect near the leading edge.

neighboring filaments meant to represent the effect of actin crosslinkers like $\alpha$-actinin, filamin or plastin (Fig 1A). The networks thus formed have effective elastic properties close to those measured for dendritic actin networks experimentally (Modeling methods and S1 Fig). Disassembly of actin filament segments is simulated as filament bond severing of aged filaments (lifetime greater than 125 $s$).

Forces by molecular motors that contribute to the generation of retrograde flow were also included. While in cells having neuronal growth cones the flow is largely due to myosin II

accumulating in the central region of the growth cone [9], treatment of XTC cells by blebbistatin, a myosin II inhibitor, does not change the speed of retrograde flow, indicating the presence of other flow motors [10]. Since the exact identity and location of these molecular motors in many cell types, including our reference XTC cells is unknown, we considered that the motor force acts in one of two modes: uniform or back, meant to represent the different spatial distributions of the retrograde flow motors. For the uniform case, a pulling force acts on every actin bead while for back pulling, a force acts only on beads further than 3.25 $\mu m$ from the leading edge. The pulling forces are external to the simulation and assumed to arise from either the lamella region (back pulling) or motors such as myosin I attached the membrane (uniform pulling) [53, 54]. The magnitude of these forces was adjusted to provide retrograde flow speeds consistent with experimental measurements [10]. We aimed for a slightly higher contributions of the leading edge force than the motor pulling force to retrograde flow, as suggested by experiments halting actin polymerization with cytochalasin D [55, 56] and discussed in section "Response to inhibition of polymerization" below.

The nascent focal adhesion is modeled at the continuum level as a capsule-shaped region of high effective viscosity located near the $z = 0$ surface of the simulation box (Fig 1E). We assume a simple frictional interaction between the actin network and focal adhesions similar to [25]. Binding and unbinding of adhesion molecules to actin can lead to biphasic and stick–slip force–velocity relations, considered in several prior modeling works such as [42, 57]. We leave; however, the investigation of such phenomena at the discrete filament level for future work. Having one focal adhesion in our 0.2 $\mu m$ wide periodic simulation box matches the typical density of nascent focal adhesions, as in Fig 7A of [10]. Here we primarily consider cell regions dominated by nascent adhesions, as in Fig 7A of [10], and do not attempt to model stress fibers around mature focal adhesions. Actin beads that are within the focal adhesion region bind and unbind from it with rate constants $k_{FA,bind}$ and $k_{FA,unbind}$, and those that are bound experience a higher viscous drag. In order to provide a scale for the focal adhesion strength, we define $\kappa_{FA}$ as the dimensionless ratio of the focal adhesion viscosity to the surrounding fluid viscosity. We vary $\kappa_{FA}$ and observe the effect it has on the retrograde flow speed and force generation within the lamellipodium.

## Actin structure and dynamics affected by presence of focal adhesion

To investigate the effect of the strength of the nascent focal adhesion on the structure and dynamics of the actin network we evolved networks to steady state (determined by a plateau in lamellipodium length, force, and concentration profiles), adjusting the pulling forces such that the retrograde flow rate is on the order of tens of nm/s, typical for XTC/fibroblast cells [10, 15] (S1 Movie). We then varied the focal adhesion strength by changing the ratio of adhesion to cytoplasmic viscosity in the range $\kappa_{FA} = 1 - 500$ (Fig 2). The lowest value, $\kappa_{FA} = 1$, is equivalent to the absence of a focal adhesion region.

The retrograde flow speed as a function of the $y$-position is shown in Fig 2A where the grey shaded box indicates the region where filaments are added and the pink box indicates the position of the nascent focal adhesion. Full color lines in Fig 2A indicate simulations run in the pull uniform mode (corresponding to uniform motor distribution) while faded lines indicate simulations run in the pull back mode (back biased motor distribution). Interestingly, the retrograde flow speeds have similar dependencies on $\kappa_{FA}$ regardless of the pull mode.

The averaged retrograde flow profile is approximately constant across most of the simulated lamellipodium (neglecting the first 0.5 $\mu m$, a region influenced by our implementation of new filament addition). As a function of $\kappa_{FA}$, the global retrograde flow speed decreases from 50 $nm/s$ to 15 $nm/s$, similar to the retrograde flow speeds measured near nascent focal adhesions

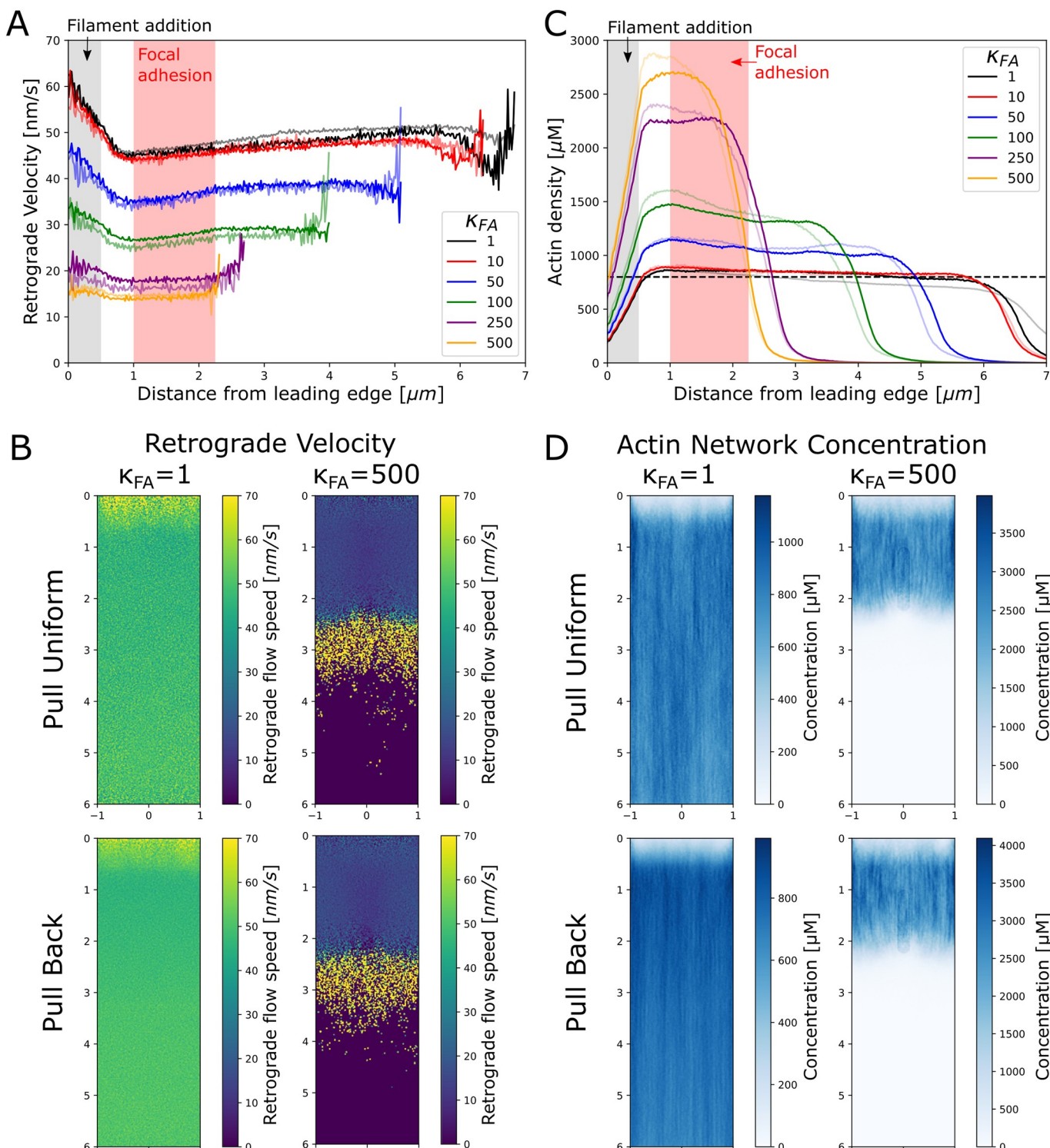

**Fig 2. Strength of nascent focal adhesion region influences flow and concentration profiles.** (A) Retrograde flow speed as a function of distance from leading edge (*y*-position) shows that the retrograde flow velocities are highest near the leading edge (gray shaded region, where added filaments compact) and flattens between this region and the red shaded region which indicates the nascent focal adhesion region. Faded (solid) lines indicate pull back (pull uniform) simulations. (B) Time-averaged retrograde flow velocity profiles are largely uniform regardless of the pull mode (uniform vs. back) with slightly higher retrograde flow speeds at the front and towards the back. (C) Actin network density as a function of distance from leading edge (*y*-position) where faded (solid) lines indicate pull back (pull uniform) mode. The actin network plateaus outside of the filament addition region (gray region), while focal adhesion strength determines the height of the

plateau value. Dashed horizontal line indicates a density of 800 $\mu M$ which is a typical value for the lamellipodium [3]. (D) Density plots of the actin network appear spatially uniform regardless of the pull mode. In correspondence with the retrograde velocity, the edges of the network have lower actin density. In B and D, the scale along the vertical ($y$) and horizontal ($x$) axes is in $\mu m$. All data calculated from simulations equilibrated for at least 300 s and averaged over at least 30 s.

in XTC cells [10]. We can thus reproduce the effect of the FA acting as a clutch, while maintaining a stable network and an approximately uniform retrograde flow speed, as observed via single molecule imaging in [10]. Between 0.5 and 1 $\mu m$, filament compaction leads to a relatively small decrease in retrograde velocity, which may be hard to detect experimentally. This decrease in retrograde flow occurs even at $\kappa_{FA} = 1$ (no adhesion), indicating that compaction can occur due to the pushing force alone without the influence of the nascent focal adhesion region. The sharp increase in retrograde flow speed magnitude and fluctuations at the very end of the lamellipodial network is due to its breakup into separate fragments of varying drag and diffusion coefficients. In cells, the region behind the lamellipodium is occupied by the lamella or the actin cortex and this increase in retrograde flow speed would be absent.

To determine if there is any local 2D spatial variation in the region around the focal adhesion (on the order of the width of the focal adhesion region, 0.25 $\mu m$), in Fig 2B we show the profile of the retrograde flow speed where the retrograde flow speed was averaged over both time and the $z$-axis. In these flow profiles there is no noticeable variation of the retrograde flow inside or around the focal adhesion region at any value of $\kappa_{FA}$, in agreement with experimental results [10]. The flow profiles suggest that the actin network in our simulations is sufficiently crosslinked to approximately behave as a uniform elastic gel [58], see Modeling methods and S1 Fig.

In order to further probe the structure of the actin network in the simulations, we plot the actin network density as a function of distance from the leading edge in Fig 2C. Full color lines in Fig 2C indicate simulations run in the pull uniform mode while faded lines indicate simulations run in the pull back mode. Filament concentrations are in the range of 800 $\mu M$ (dashed line), which is the typical value for the density of the lamellipodial actin network [3]. Behind the filament addition region, the density of the actin network nearly plateaus, as anticipated from the nearly spatially uniform retrograde flow (Fig 2A). As $\kappa_{FA}$ increases, the plateau value increases (since filaments are always added at the same rate and retrograde flow speed decreases), leading to a decrease in overall network length (since the actin segment lifetime remains fixed). To determine if there is a local density increase in front of the focal adhesion region, we plot the time average F-actin density along $x$ and $y$ in Fig 2D. There is barely any density increase near the focal adhesion, in line with the velocity measurements of Fig 2B. At the highest $\kappa_{FA}$ value of 500, there is a slight increase in density at the focal adhesion boundary, but this is due to our implementation of the focal adhesion region that introduces a slight attractive force on actin beads.

## Force distribution affected by focal adhesion

While the local network velocity and density are not significantly perturbed by the presence of a nascent focal adhesion (Fig 2B and 2D), the local force balance within the actin network is affected by the presence of a nascent focal adhesion as shown in Fig 3A. Here we calculate the average tension force within actin filament segments and categorize it as either compressive if the bond force is positive (indicated by red in Fig 3A) or as tensile if the force of the bond is negative (indicated by blue in Fig 3A). The sign of the tension switches in the lamellipodium where the region of the network closest to the leading edge is under compression while past the focal adhesion region (for large enough $\kappa_{FA}$) the network is under extension. Interestingly,

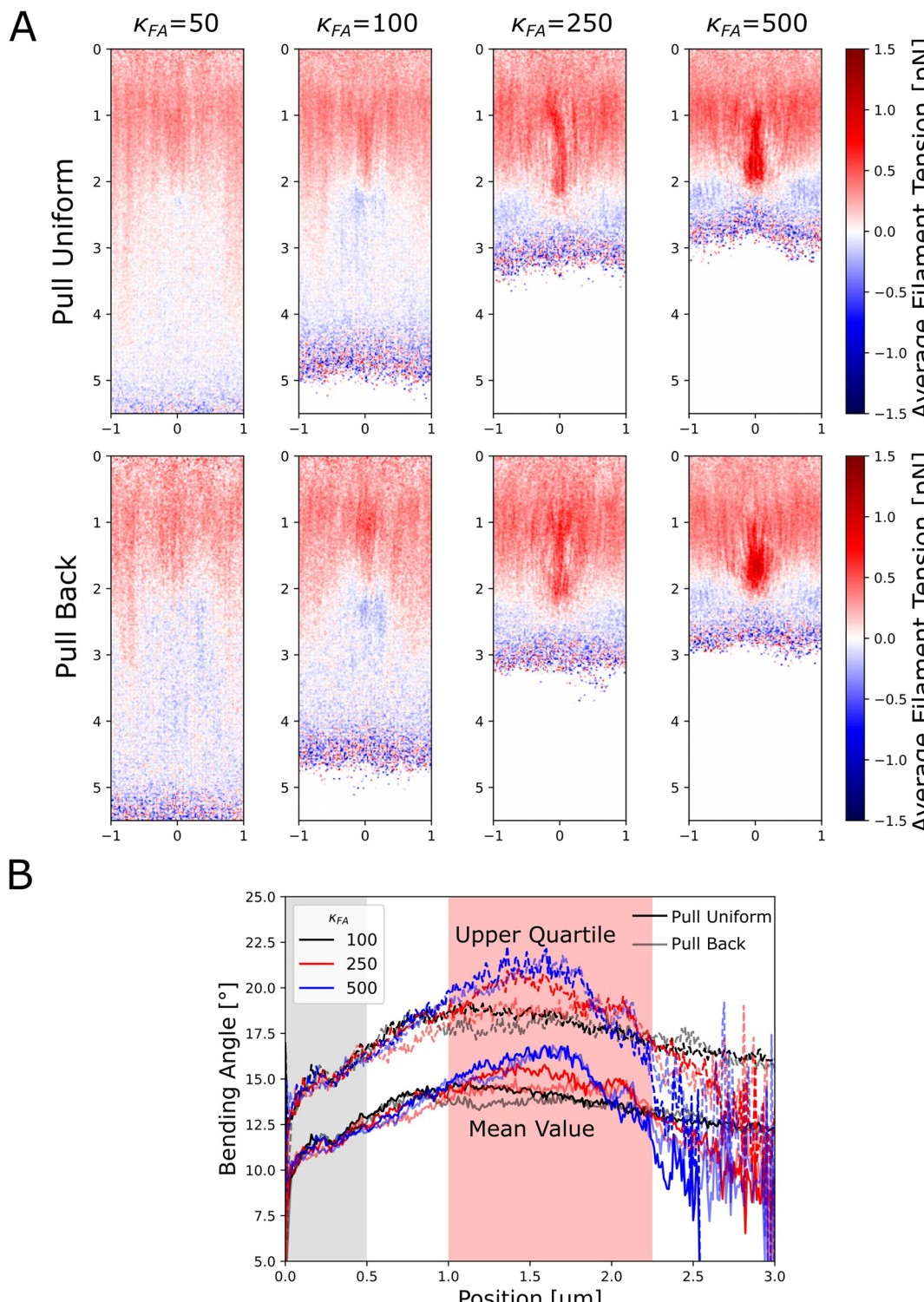

**Fig 3. The simulated actin network is under compression in front of the focal adhesion region and under extension behind the focal adhesion region.** (A) The filament tension (averaged over both time and the *z* direction at steady state) indicates that compression occurs in front and inside the focal adhesion region (indicated by red) while a smaller amount of extension occurs below this region (indicated by blue). The amount of compression increases as the focal adhesion strength increases and is maximal within the focal adhesion region. There is no significant difference between the two pulling modes. The scale along the vertical (*y*) and horizontal (*x*) axes is in *μm*. (B) The bending angles between filament segments of length

0.5 $\mu m$ (averaged over both time and the $x$, $z$ directions at steady state) show an increase within the focal adhesion region dependent on higher focal adhesion strengths. At the highest focal adhesion strengths investigated, the bending angle reaches a peak near the middle of the focal adhesion region. Solid (faded) lines indicate uniform (back) pulling. Continuous (dotted) lines show mean (upper quartile) values. All data calculated from simulations equilibrated for at least 300 s and averaged over at least 30 s.

this switch in tension sign occurs for both pulling modes, and marginally even when the back pulling force is not acting on the network since the network at high $\kappa_{FA}$ does not extend into the back pulling region (see pull back case, $\kappa_{FA} = 500$). The forces in the network increase in strength as $\kappa_{FA}$ increases, further indicating that the focal adhesion contributes to the switch from compression to extension (while for $\kappa_{FA} = 1$ the network is nearly completely under compressive stress; not shown).

Since the large amounts of compressive strain may lead to bending of filaments that could be important for filament severing [59], we measured the average bending angle between actin filament segments at the ends of 1 $\mu m$ long filaments as a function of distance from the leading edge to determine if there was filament buckling (Fig 3B). Both pulling modes provided similar curves. The bending angle increases to a maximum in front of the focal adhesion for $\kappa_{FA} = 100$, while the bending is larger and occurs in the middle of the focal adhesion region for the highest $\kappa_{FA}$ values investigated. The bending angles of the filaments in the upper quartile reach values close to 30 degrees, at which cofilin is expected to enhance severing [59]. These data indicate that strong focal adhesions may lead to bending-induced severing and possible reconfiguration of the actin network (which was not included in these simulations).

Steady state in our simulations is reached through a balance of forces (Fig 4). To investigate this balance, we quantified the viscous and pushing/pulling forces in the network as a function of the focal adhesion strength. Forces were normalized either to force per $\mu$m of the leading edge or to force per actin bead in the simulation.

As $\kappa_{FA}$ increases, the drag force due to the focal adhesion region increases while the drag due to the fluid cytoplasm decreases, for both pulling modes (Fig 4A). The magnitude of the pushing (polymerization) and pulling (motor) forces on the network are also measured (Fig 4B). As the focal adhesion strength increases, the pushing force increases because there are more filaments in the pushing region and filaments are inserted at a fixed constant rate. For uniform pulling, the pull force per bead is by definition constant irrespective of the focal adhesion strength. In contrast, in the pull back case, the pulling force decreases with increasing focal adhesion strength since there are fewer actin beads in the pull back region.

A leading edge pushing force versus velocity curve can also be constructed for our model (Fig 4C). We note that this curve is dependent on the simplified assumptions of our model, including a fixed polymerization (i.e. filament insertion) rate while in reality a reduction of polymerization with force is expected [50, 58]. Whether the relationship between the external force at the leading edge the actin network extension speed should be concave or convex for actin networks has been debated [40, 41, 49, 60]. Our simulations result in a convex curve for the retrograde flow speed as a function of the pushing force on the actin network.

## Response to inhibition of polymerization

A common experimental perturbation to lamellipodia dynamics is the addition of drugs that inhibit polymerization. Experimentally, addition of cytochalasin D leads to detachment of the actin network from the membrane, a reduction of retrograde flow, and eventual network disassembly [55, 56]. Such experiments reveal the relative contribution of pushing from the leading edge and motor pulling on the retrograde flow. We reproduce the effect of cytochalasin D

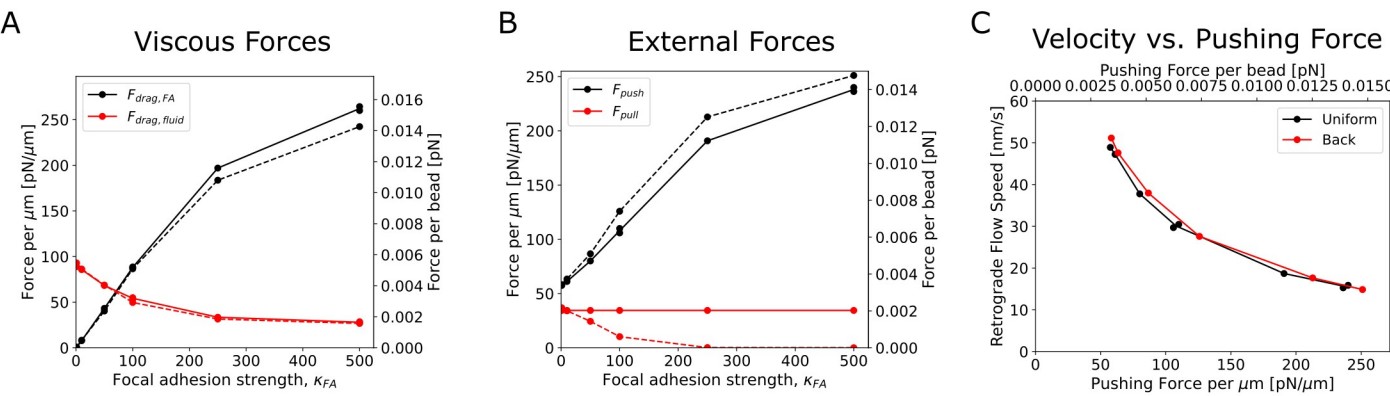

**Fig 4. Force balance in the actin network changes as the focal adhesion strength increases.** Forces in these plots are the net force on the actin network divided by either the leading edge width of 2 $\mu m$ (force per $\mu m$) or per number of actin beads in the simulation (force per bead). (A) Viscous forces versus focal adhesion strength. Drag due to the focal adhesion region increases as focal adhesion strength increases while that due to the surrounding fluid decreases for both pull uniform (solid lines) and pull back (dashed lines) modes. (B) External forces versus focal adhesion strength. As the focal adhesion strength increases the pushing force increases and, for the pull uniform case, the pulling force decreases to zero. The pulling force per bead is, by definition, fixed in the uniform case so it remains constant. (C) Retrograde flow speed as a function of the external pushing force. Higher pushing forces and lower retrograde flow speeds correspond to higher focal adhesion strengths. Plot shows a decreasing, convex curve. In all panels, forces are shown as per bead in the lamellipodium or as force per $\mu m$ along the leading edge. To check reproducibility, replicates were run for $\kappa_{FA}$ = 100 and 500 in the pull uniform case and shown with separate data points in all panels; little variance was seen in the forces or retrograde flow speed between the two runs. All data calculated from simulations equilibrated for at least 225 s and averaged over at least 30 s.

addition by ceasing addition of filaments and membrane pushing forces at the leading edge after steady state is reached. In these simulations on the effects of cytochalasin D, the retrograde flow speed decreases to approximately 55% of the initial value when under back pulling and to 45% under uniform pulling at $\kappa_{FA}$ = 1 (Fig 5A). As the focal adhesion strength increases, the retrograde flow decreases by more than 50% after ceasing filament addition, suggesting that the pushing force in these cases was larger compared to the pulling force at steady state (as shown in Fig 4B). This dependence of the degree of reduction of retrograde flow on adhesion strength could explain why the retrograde flow was reduced to different levels after cytochalasin D treatment in prior experimental studies [55, 56]. Images of the lamellipodial network show that the network retreats both from the leading edge and the back of the network over time due to elimination of addition of filaments at the front and the gradual removal of filaments at the back due to aging (Fig 5B and S2 Movie). Filament disassembly did not result in enhanced retrograde flow, unlike the mechanism proposed in [61].

## Arc formation at the lamellipodium boundary

Having generated a steady state filament-level simulation, we next examined if our simulations can be applied to model other common features associated with lamellipodia, which depend on changes of filament distribution. One such phenomenon is the bundling of filaments into arc-like structures near the back of the lamellipodium, at the boundary with the lamella region [56, 62–64]. The formation of these arcs near mature focal adhesions has also been seen in a continuum gel model used to describe the formation of an arc geometry [32].

To investigate the reorganization of the actin network at the back of the lamellipodium, we moved the focal adhesion further back to start at $y$ = 2.375 $\mu m$ (Fig 6) and used a $\kappa_{FA}$ = 250 that provides a relatively large resistance to motion. For this set of simulations we used the uniform pulling mechanism.

We found that actin bundle arcs can form under these conditions by a combination of dynamic crosslinkers and long-lived crosslinkers meant to represent the Arp 2/3 complex with

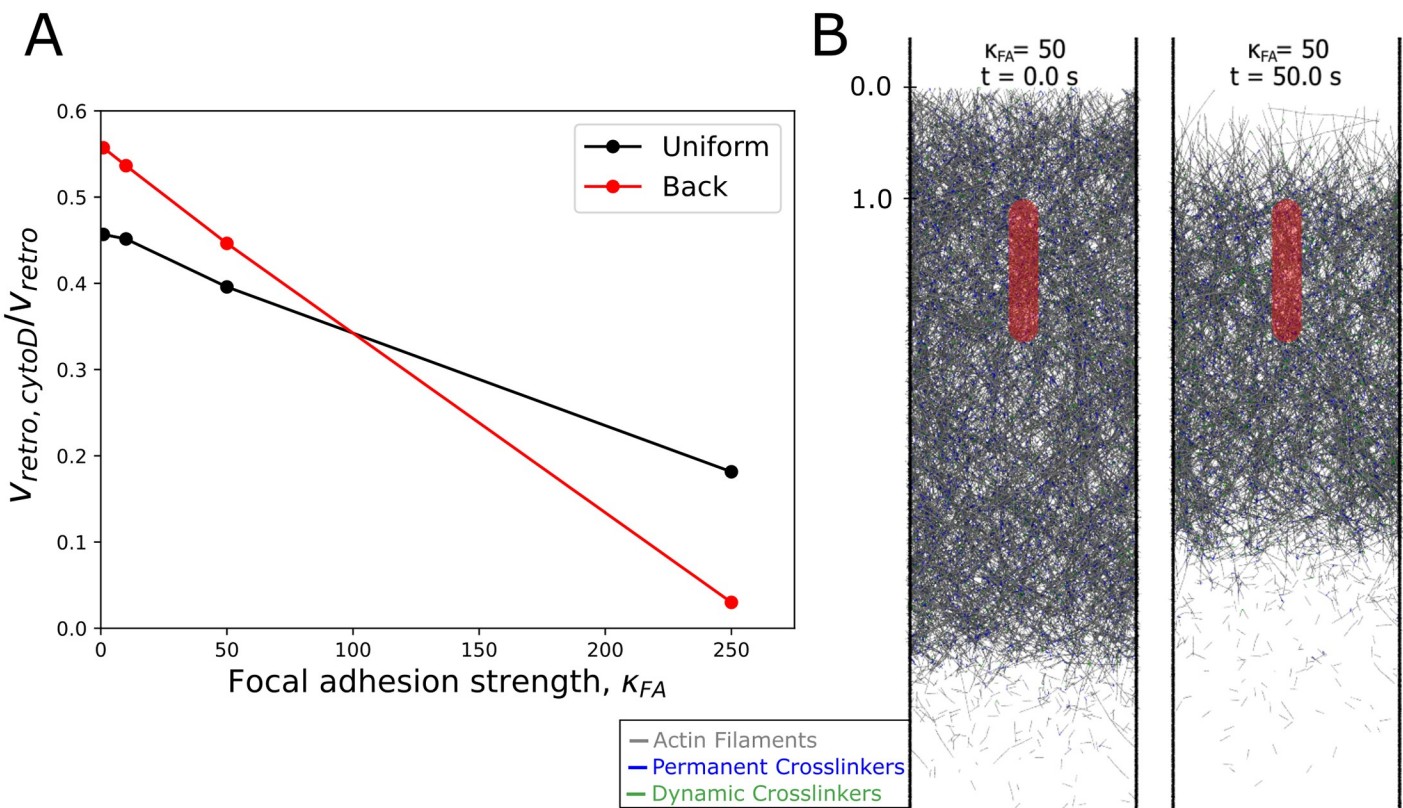

**Fig 5. Simulated effect of cytochalasin D on retrograde flow speed captures experimental results.** (A) Effect of addition of cytochalasin D was modeled as stopping addition of filaments after reaching steady state. The retrograde flow speed $v_{retro,cytoD}$ was measured between 30–35 $s$ after stopping filament addition and compared to the prior retrograde flow speed at steady state, $v_{retro}$, as a function of focal adhesion strength. Both velocities are averages over all filament beads. (B) Representative snapshots of the simulated effect of cytochalasin D addition at the time of addition ($t = 0$) and at $t = 50$ $s$ from a pull back simulation. The perturbation results in retreat of the actin network from both the leading edge due to retrograde flow without addition of filaments and from the back due to filament disassembly. The scale along the vertical ($y$) axis is in $\mu m$. Blue (green) line segments represent permanent (dynamic) crosslinkers.

a finite rate of debranching. We first investigated how the concentration of dynamic crosslinkers $C_{dynamic}$ (at 1, 10, and 30 $\mu M$) affects the network, keeping the long-lived crosslinkers unchanged compared to Figs 2–5 (Fig 6A and S3 Movie). Under steady state conditions, despite transient density inhomogeneities at low $C_{dynamic}$, and local bundling at high $C_{dynamic}$, the network maintained an approximately uniform concentration through the focal adhesion region (Fig 6A). In contrast, if we allow the permanent crosslinkers to be removed at a constant rate with an average lifetime of 20 $s$, the network rearranges close to the focal adhesion region at intermediate $C_{dynamic}$ (Fig 6B and S4 Movie). In the intermediate dynamic crosslinker concentration case there is strong aggregation of the filaments into arcs in front of the focal adhesion region and there is a clear peak in the actin density. The profile of filament tension also changes depending on $C_{dynamic}$ (S2 Fig): filaments transition from compression to extension at $C_{dynamic} = 1$ $\mu M$ while they are mostly compressed at 10 and 30 $\mu M$. There is little filament stretching or compression within the arc bundle itself.

We also ran the same simulations of Fig 6 in the pull back mode (S3 Fig). An actin bundle arc still forms at the back of the lamellipodium at $C_{dynamic} = 10$ $\mu M$ with long-lived crosslinkers. However, the bundle is less well defined when comparing Fig 6B with S3(B) Fig: the pull back force acts more strongly on the back half of the lamellipodium, disrupting tight bundle

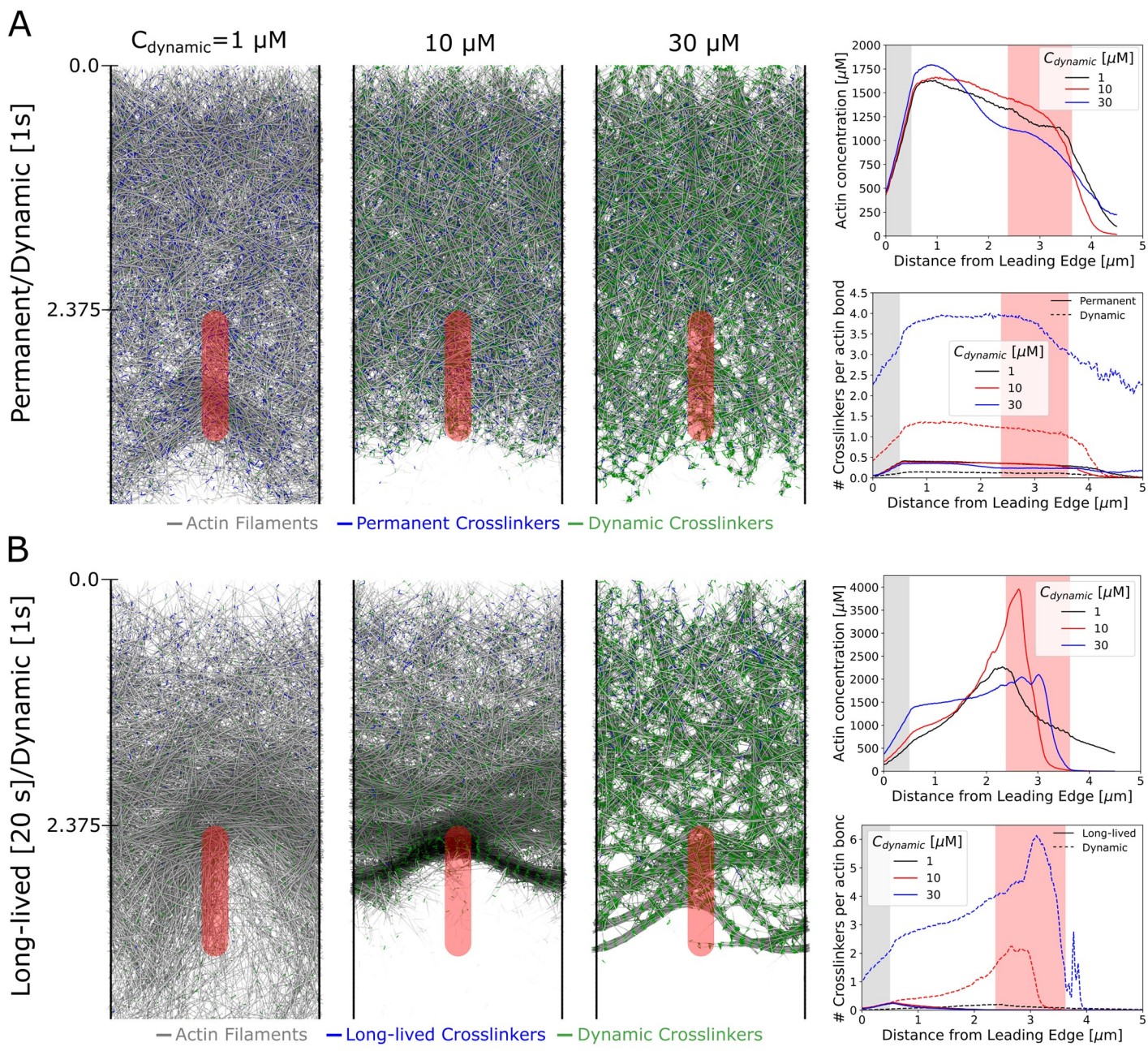

**Fig 6. Actin bundle arc formation at the lamellipodium boundary.** Filament rearrangement leading to bundling of the actin filaments in the focal adhesion region can occur but only if the restrictions due to the permanent crosslinkers are relieved over time. A mature focal adhesion at the back of the lamellipodium is simulated by starting the focal adhesion region at $y = 2.375 \ \mu m$ and using a focal adhesion strength $\kappa_{FA} = 250$. For these simulations we used the uniform pulling mechanism. (A) Snapshots (left) and concentration profiles (right) by a combination of permanent and dynamics crosslinkers. Increasing the number of dynamic crosslinkers results in tighter bundles of filaments while leaving the overall spatial concentration of filaments relatively unchanged and there is no bundling of filaments near the focal adhesion region. Blue (green) line segments represent permanent (dynamic) crosslinkers. (B) Same as panel A but replacing permanent crosslinkers with crosslinkers that have a finite lifetime of 20 $s$. Bundling, together with a peak in actin concentration, occurs in front of the focal adhesion region at intermediate dynamic crosslinker concentrations. In the simulation snapshots, the scale along the vertical ($y$) axis is in $\mu m$. Blue (green) line segments represent long-lived (dynamic) crosslinkers. All data calculated from simulations equilibrated for at least 140 s and averaged over at least 35 s.

formation in the focal adhesion region. This is an instance where the pull uniform and pull back modes lead to differences in lamellipodium morphology.

The results of Fig 6B show how actin bundles may form through the reorganization of the branched network at the back of the lamellipodium [62, 64] (for example through debranching [13, 65], distributed turnover [66, 67], and dynamic crosslinking). A related mechanism for arc formation observed experimentally involves collapse of microspike bundles into arcs at the lamellipodium-lamellum interface, even when myosin-II is inhibited by blebbistatin [68, 69]. However, we note that the proposed mechanism of filament alignment and compaction may act in conjunction with (or help trigger) myosin-II to form transverse arcs [12, 62, 63] and the two mechanisms do not need to be mutually exclusive. Though we do not simulate discrete myosins, the dynamic crosslinkers in our simulations may be playing a similar bundling role. Bundle formation through myosin II contraction has previously been modeled at the discrete filament level [70], including related models of actomyosin bundle formation in cytokinesis [71, 72]. Our model shows how bundling can also result from the forces of retrograde flow, focal adhesion geometry, and crosslinking.

## Microspike formation within lamellipodia

Other common bundle structures forming within the lamellipodia are filopodia (bundles that protrude beyond the leading edge) and microspikes (bundles embedded in the lamellipodium) [19]. Within our current simulation with a fixed leading edge, we explored the possibility of generating microspike bundles. Microspikes have been demonstrated to form through Ena/VASP-mediated polymerization at the leading edge [19]. Already, bundled actin reminiscent of microspikes can be seen in the high dynamic cross-linking case ($C_{dynamic}$ = 30 $\mu M$) of Fig 6. However, the bundles of Fig 6 are oriented along multiple directions instead of being oriented primarily along the direction of retrograde flow as is typical of microspikes.

We performed simulations to determine if microspike bundles, with orientations similar to those seen in experiments, can form and remain stable in our simulations. To represent the effect of Ena/VASP at the leading edge, we altered the manner in which filaments were added at the leading edge (Fig 7A). Half of the filaments were added in the same manner as in Figs 2–6, while half are added with a vertical orientation, i.e. with their axes along $y$ (Fig 7A). We further investigated two different modes of vertical orientation addition: adding vertical filaments near existing filaments (pointed end of the newly added filament placed 35 $nm$ away from an existing filament) or adding vertical filaments with a random $x$-coordinate (Fig 7B).

The resulting networks are shown in Fig 7C and S5 Movie, where microspike-like bundles can be seen in either mode of vertical filament addition, though they are better defined when vertical filaments are added near existing filaments. Similar to experiments (see for example S3 Movie in [45]), the bundles are long-lived as they extend throughout the entire lamellipodium to the boundary where the network disassembles. Similar to microspikes and filopodia in experiments [18, 45, 68, 69], they merge with each other to form inverted Y structures, as seen more frequently in the random addition case.

The simulations of Fig 7 and S5 Movie indicate that changing the angle at which filaments are introduced at the leading edge can have persistent effects on the network topology in the presence of actin filament crosslinkers. Our model thus suggests that microspikes (and possibly filopodia) may form by small changes in filament orientation at the leading edge. Indeed, the two proposed competing models of filopodia formation, convergent elongation versus de novo nucleation make different assumptions about how this change of orientation comes about [18, 19, 73]. We have not implemented sufficient molecular resolution in our model to resolve between these two mechanisms; however, the model in Fig 7 can be considered an

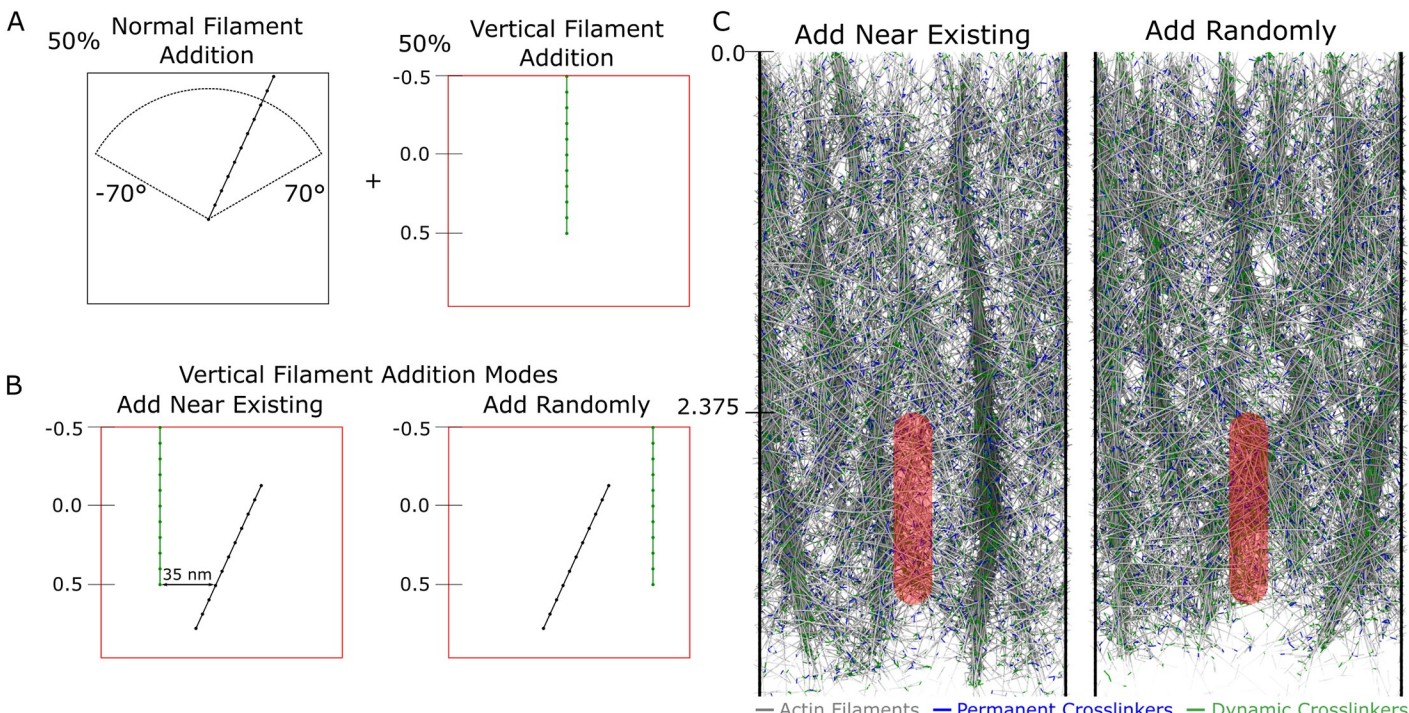

**Fig 7. Vertical filament bundles reminiscent of microspikes form if the method of filament addition is modified.** Retrograde flow simulations show microspike-like bundles when focal adhesion with $\kappa_{FA}$ = 250 with top edge at $y$ = 2.375 $\mu m$, and uniform pulling is applied, as in Fig 6. The concentration of dynamic crosslinkers was fixed at 10 $\mu M$. (A) Schematic of vertical filament addition mode: half of filaments added initially have a vertical orientation while the remaining half are added with initial orientation between -70˚ and 70˚ as shown in prior figures. (B) Vertical addition occurs with filaments either added with their pointed ends 35 $nm$ away from existing filaments along the $x$ direction, or at a random $x$ position. (C) Steady state simulation snapshots. Vertical filament addition either near existing filaments or randomly results in vertical filament bundles similar to microspikes, with the "near existing" filament addition mode resulting in tighter bundles. The scale along the vertical ($y$) axis is in $\mu m$. Blue (green) line segments represent permanent (dynamic) crosslinkers. Simulation snapshots taken after 275 s of simulation time.

intermediate between the two mechanisms: vertical filament addition is a de novo mechanism (though not necessarily coordinated along the $x$ leading edge direction), while bundling with pre-existing filaments has features similar to convergent elongation.

The formation of bundles within our simulated networks in both Figs 6 and 7 is reminiscent of actomyosin bundle formation within a disordered cortical actin network [74, 75]. While we do not explicitly model myosin motors, our simulations show how dynamic or permanent passive crosslinkers, in combination with actin turnover and alignment through flow (Fig 6) or polymerization (Fig 7) can promote and maintain bundled structures within disordered networks.

## Robustness of model to variations

As a test of the robustness of our model, as well as a preliminary investigation into future improvements, we performed additional simulations where we change an aspect of the model and investigate the effect on retrograde flow, network density, and force transmission within the network.

To investigate how our reference simulations depend on the choice of distance between actin beads ($l_0$ = 100 $nm$ or 37 actin subunits), we reduced the spacing between to $l_0$ = 48.6 $nm$ (18 actin subunits). This value is close to the spacing between branch points in fibroblast lamellipodia that occur predominately at intervals of 36 $nm$ along mother filaments (equal to their helical pitch) [15], within 1 $\mu m$ to the leading edge. To compensate for the higher number

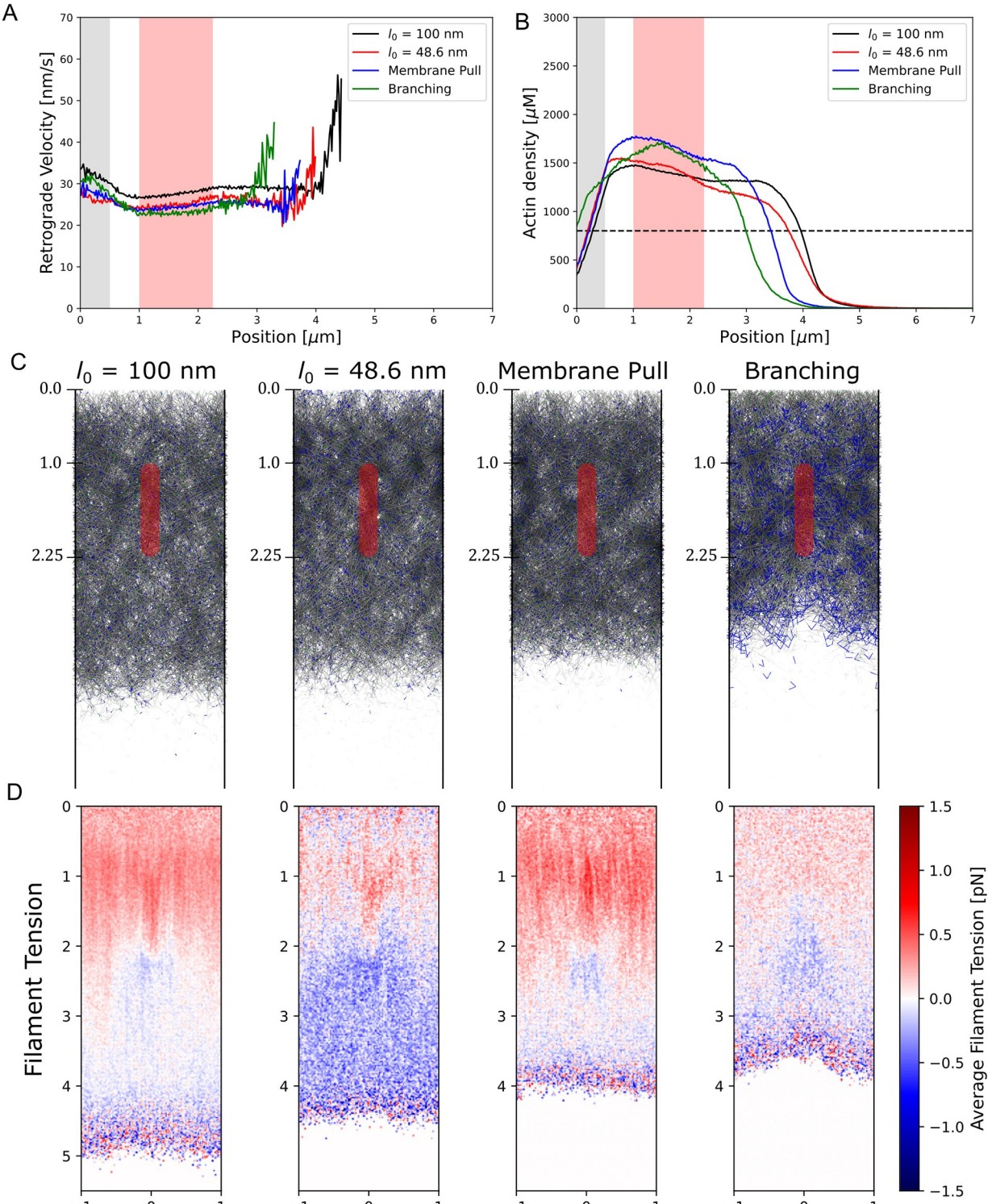

**Fig 8. Simulations investigating model modifications.** Simulations investigating the effect of reducing the actin bead spacing by approximately half, exerting motor forces only on actin beads within the bottom tenth of the simulation box, and implementing branches instead of permanent crosslinkers to represent the Arp2/3 complex. Simulations were run with $\kappa_{FA} = 100$, focal adhesion starting at $y = 1\ \mu m$, and, except for the membrane pull simulations, the pull uniform mode. (A) Retrograde flow plot as a function of distance from the leading edge for the three modified simulations show qualitative similarities to the reference simulations at $l_0 = 100\ nm$. Grey region indicates the region over which filaments are added while the red region indicates the focal adhesion position (B) Actin density as a function of distance from the leading edge

again shows similar behavior among the four simulations. (C) Snapshots of the network after reaching steady state with actin in grey, permanent (dynamic) crosslinkers in blue (green). Branching simulations have both the actin filament segments on the mother and daughter filaments that compose the branch colored in blue. Each simulation was run for at least 190 s and averaged over at least 20 s. (D) Filament tension plots of the four simulations where compression is colored in red and extension in blue.

of filament beads in the simulation, we reduced the drag coefficients and pulling and pushing force per bead by half. To compare to our reference simulations, we measured the retrograde flow speed, density, and the compression/extension of filament bonds in the network with uniform pulling at $\kappa_{FA} = 100$. The two cases only have minor differences (Fig 8). The retrograde flow speed is slightly lower than in the reference simulations (Fig 8A), leading to a slightly higher actin density in front of the focal adhesion, but a lower density behind the focal adhesion (Fig 8B). In the $l_0 = 48.6$ $nm$ simulations the filament tension is higher behind the focal adhesion than in the $l_0 = 100$ $nm$ simulations (Fig 8D). This higher level of extension may be a result of the higher cross-linking opportunities during the filament insertion process, which leads to $\sim 20\%$ more permanent crosslinkers over the $l_0 = 100$ $nm$ simulations.

Next, we investigated how our uniform pulling assumption affects our results. Motors that contribute to retrograde flow could associate with the plasma membrane such as certain myosin I motors [53, 54]. We investigated this possibility ("membrane pulling") in more detail by only pulling actin beads within 20 nm of the bottom of the simulation box (representing the membrane adhered to the external substrate), outside the focal adhesion region (see Modeling methods section). Regardless of these changes, these simulations result in comparable values of the retrograde flow, network density, and average filament tension to the pull uniform simulations at the same focal adhesion strength (Fig 8A, 8B and 8D). In addition to these three numerical parameters, the structure of the lamellipodium is also largely similar between the pull uniform and membrane pulling cases (Fig 8C).

Finally, we investigate how our method of adding permanent crosslinkers affects our simulation results, when compared to branching by the Arp2/3 complex that maintains a branch angle of $\sim 70°$. For details on the branching implementation see Modeling methods and S4 Fig. The retrograde flow at the same filament addition rate and focal adhesion strength, $\kappa_{FA}$, is similar to simulations with permanent crosslinkers, but with a higher speed near $y = 0$ and a lower speed further away (Fig 8A). The filament density peaks near the middle of the FA region (Fig 8B). The force distribution within a network generated by branches is also similar to that with permanent crosslinkers (Fig 8D) where compression is seen near the leading edge and extension near the back of the focal adhesion region. Overall, a network composed of branches behaves similarly to a network with permanent crosslinkers, at least as far as the retrograde flow, density, and force profiles are concerned.

## Discussion

We simulated a lamellipodial actin network at the level of individual filaments, undergoing retrograde flow at steady state in the presence of a focal adhesion, in order to investigate the forces and deformations within the lamellipodium. We assumed that the leading edge remained at a fixed position, as would be the case for lamellipodia of a stationary cell, and implemented forces exerted by polymerization at the leading edge as well as simplified external motor-based pulling forces. The presence of a nascent focal adhesion, represented as a region of enhanced friction, does not locally perturb retrograde flow, in agreement with the reference experiments [10]. Increasing the strength of the focal adhesion leads to several changes including reducing the overall retrograde flow speed, in agreement with the molecular clutch hypothesis. Increasing focal adhesion strength also enhances compression in front of the focal

adhesion region, and bending of filaments in front and interior to the focal adhesion region that could be regions of the lamellipodium prone to severing. As a test of the force balance in the model, we simulated the effects of addition of cytochalasin D which lead to a reduction in retrograde flow by 50–100%, as seen in prior experiments. Finally, we found that bundles of actin can form in our simulations, reminiscent of transverse arcs or microspikes/filopodia, when modifying either the lifetime of long-lived/permanent crosslinkers (Arp2/3 complex) or the method of filament addition, hinting at mechanisms for how these structures may easily form by small changes in actin filament crosslinker and polymerization proteins.

This work highlights how lamellipodia may experience compressive or tensile stresses depending on distance to the leading edge, location and strength of the focal adhesion, and cross-linker concentration. Polymerization near the leading edge generally leads to compression, however simulated motor pulling can also create tensile stress in the back region of the lamellipodium, behind nascent adhesions (Fig 1 and S2 Fig). To our knowledge, the possibility of such a tensile-stress dominated region at the back of the lamellipodium, ahead of the myosin II-dominated contractile lamella/cortical region has not been considered in prior models. However, such changes from compression to extension as function to the distance from the leading edge may indicate different mechanical requirements across the lamellipodium, regulating the cycle of focal adhesion maturation and turnover.

The mechanics of lamellipodia of fast-moving keratocyte cells have been considered in several prior modeling studies, though not at the single filament level [76]. Unlike the stationary lamellipodia that we considered, keratocytes are motile, have low retrograde flow speed, few focal adhesions, a larger lamellipodium that spans the cell body, and highest traction forces at the rear of the cell [76, 77]. Interestingly, blebbistatin treatment of keratocytes decreased the lifetime of actin speckles [78], which may indicate that the keratocyte actin network is largely under tensile stresses which, in the presence of cofilin, have been found to reduce severing under certain conditions in vitro [79] (though not by other groups [80, 81]). A change from compression to extension with increasing distance from the leading edge could indicate why lamellipodia in electron micrographs [12] show a transition from short, branched filaments at the leading edge, able to support compression without significant buckling, to longer, less branched filaments deeper in the lamellipodium, able to support extension. We note that some studies concluded that the keratocyte lamellipodium body is under myosin II-driven compression, except for the network within a $\mu m$ of the leading edge that is under extension [48, 82]. Tensile stresses at the leading edge, however, would imply that polymerization is not exerting a pushing force on the membrane.

The above remarks indicate how mechanical models of motile cells may require extending the lamellipodium model to account for structural network changes (for example through filament severing, debranching, and annealing [47, 83]), distinct lamellipodial and lamellar/cortical networks [56], as well as cytoskeletal and adhesive properties over the whole cell [84]. Such approaches would address phenomena beyond the scope of this work, including the biphasic behavior of traction stress versus retrograde flow through the lamellipodium/lamella region [24, 25] and cell velocity versus adhesive strength [84].

The extended regions of compressive force we observe in some of our simulations with adhesions placed near the back of the lamellipodium, for example in case two of S2(B) Fig, could lead to buckling of the network as also seen in our mechanical compression simulations (S1(D) Fig, case without slab boundary confinement). Indeed, fibroblast lamellipodia have previously been seen to buckle due to the force from membrane tension when transitioning from processive to stationary motion on a fibronectin coated coverslip, suggesting there are large compressive forces near the leading edge during certain periods of the protrusion-retraction

cycle [85, 86]. Buckling has also been seen perpendicular to the leading edge in keratocytes [87].

Having established a steady-state model at the filament level allows for future improvements and refinement to account for important regulatory mechanisms closer to the molecular level. In Fig 8 we found that many aspects of the model are robust to changes of actin filament discretization, mode of uniform pulling, and branching mechanisms. Future work could include a more detailed description of force-dependent polymerization at the leading edge as has been done in several other models [36, 39, 41, 49, 50, 58]. Motors could also be explicitly modeled as in prior works [70, 88] allowing for significantly more informative force profiles and network morphology changes due to pulling than in our current model. The identity of these motors that contribute to retrograde flow is unknown, but several myosin I motors connect actin to the membrane which may allow for force to be exerted on the membrane and pull actin in the retrograde direction [53, 54]. Addition of a membrane that responds to forces from the lamellipodial network is also possible, to model protrusion and other non-steady state conditions. The model for the focal adhesion could also be improved by allowing it to change in size/shape in response to forces exerted on it by the actin network. One method of implementing a more complex focal adhesion is to incorporate discrete talin and vinculin molecules that are recruited to the focal adhesion region as a function of increased force on the focal adhesion.

In the simulations presented in this work, the lamellipodial network is extensively cross-linked and behaves as a single, cohesive network with relatively little local perturbation to the retrograde flow near the nascent focal adhesion region even at the highest focal adhesion strengths investigated (Fig 2B). Since mature focal adhesions locally alter flow [10], this suggests that the lamellipodial networks in these regions may undergo significant remodeling, possibly due to destabilization [89] or bending and severing by cofilin [20]. Our model could be extended to allow filaments that are under significant bending stress to have a higher probability of severing to observe this behavior.

The model developed in this paper allowed us to investigate the force balance within a simulated steady-state lamellipodium at the filament level including the effects of crosslinkers, leading edge forces, pulling forces, and forces from the focal adhesion in order to find regions of compression/extension and modes of bundle formation within the lamellipodium. As mentioned above, several improvements to the model could allow for more detailed investigations to advance the understanding of how the lamellipodium forms and generates force. We plan to incorporate these improvements in future implementations of the model.

## Modeling methods

We simulated a stationary lamellipodium within a rectangular simulation box that has periodic boundary conditions along the $x$ direction with a width of 2 $\mu$m, has no length limit along the $y$ direction, and has a height 0.2 $\mu m$ in the $z$ direction. The direction of the retrograde flow in our simulations is along the positive $y$ direction (towards the cell center) with the position of the leading edge being at $y = 0$ as described below.

### Filament representation

In our Brownian dynamics simulations (Fig 1) actin filaments are modeled as discrete beads connected via springs with each filament segment (bead-spring-bead) representing 37 actin monomers [90, 91]. Forces in our simulations can be divided into four groups: intra-filament forces, inter-filament forces, external forces, and thermal forces. Intra-filament forces include spring forces acting between each pair of adjacent beads and bending forces acting between

each triplet of neighboring beads. Inter-filament forces include crosslinker spring forces, which act between specified pairs of beads, and excluded volume forces which act between unique actin filament segments. External forces, which act on individual actin beads, include a pushing force representing force by polymerization at the leading edge, a pulling force representing motors, and a (smoothed) hard wall boundary force acting to restrict beads in the z-direction. The sum of forces on each bead $i$ is used to evolve the positions of the bead, $\mathbf{r}_i$, over time via 3D Brownian dynamics according to the following equation:

$$\zeta_b d\mathbf{r}_i/dt = \mathbf{F}_i^{spring} + \mathbf{F}_i^{bend} + \mathbf{F}_i^{crosslink} + \mathbf{F}_i^{excluded} + \mathbf{F}_i^{leading} + \mathbf{F}_i^{motor} + \mathbf{F}_i^{thermal}, \tag{1}$$

where $\zeta_b$ is the drag coefficient for a cylindrical segment averaged over both the long and short axes of the cylinder. Specifically, $\zeta_b = 4\pi\eta_{fluid}\, l_0/[\ln(l_0/d) + 0.84]$ where $\eta_{fluid}$ is the viscosity of the cytoplasmic environment, $l_0 = 0.1\ \mu m$ is the length of the cylindrical segment, and $d$ is the diameter of the cylindrical segment (7.0 $nm$) [72]. We used a viscosity of $\eta_{fluid} = 0.3\ Pa \cdot s$ which is approximately 350 times that of water [90].

## Intra-filament forces

The spring force, indicated in Fig 1B, can be written as [72, 91]:

$$\mathbf{F}_i^{spring} = -k^{actin}(d_{ij} - l_0)\hat{d}_{ij} \tag{2}$$

where $k^{actin}$ is the bond spring constant used between actin beads, $l_0 = 0.1\ \mu m$ is the equilibrium bond separation, $d_{ij} = |\mathbf{r}_i - \mathbf{r}_i|$, and $\hat{d}_{ij}$ is the unit vector from bead $j$ to bead $i$. For the majority of simulations we used $k^{actin} = 10^3\ pN/\mu m$, which is a higher than the value used in [90, 91] but smaller than $k^{actin} = 1.69 \cdot 10^4\ pN/\mu m$ in [27]. All these values are smaller than the value derived by considering the elastic modulus of single actin filaments [72], to allow for a larger simulation time step at the expense of actin filaments being more easily extendable. The implications of the $k^{actin}$ value is discussed in section "Mechanical characterization of simulated lamellipodial network" below. The bending force can be written as

$$\mathbf{F}_i^{bend} = \kappa/l_0 \sum_{j=2}^{N-1} \frac{\partial(\hat{d}_{(j+1)j} \cdot \hat{d}_{j(j-1)})}{\partial \mathbf{r}_i} \tag{3}$$

where $\kappa = k_B T l_p$ is the flexural rigidity and $l_p = 17\ \mu m$ is the persistence length of a filament (we use a value somewhat higher than 9.8 $\mu m$ of bare actin and 2.2 $\mu m$ of cofilin-actin) [59, 92–94].

## Crosslinker, excluded volume, and confining forces

Crosslinkers in our simulations act on actin filaments to represent Arp2/3 complex, $\alpha$-actinin, filamin or plastin. They are simulated as springs connecting beads of actin filaments without bending stiffness that some prior works included [27, 29] and are only allowed to bind to actin beads rather than arbitrary locations along the filament [27, 29, 88]. We allow crosslinkers to form or break over time as detailed in section "Network generation, crosslinking, and disassembly" below. Crosslinker spring forces are calculated in the same manner as filament spring forces in Eq 2, but with a softer spring constant $k^{crosslink} = 100\ pN/\mu m$ and an equilibrium length of $l_0^{crosslink} = 35\ nm$, a distance that we picked to be shorter than the filament segment length $l_0$, and of order the length of $\alpha$-actinin [95].

Excluded volume forces act when two filament segments, $\alpha$ and $\beta$, overlap, and are exerted along the direction of vector $\mathbf{d}_{\alpha\beta}$ which joins the closest approach points between the two

segments, as in [27] and indicated in Fig 1D. The excluded volume force is defined as $\mathbf{F}_\alpha^{excluded} = -k^{excluded}(|\mathbf{d}_{\alpha\beta}| - d^{excluded})\hat{d}_{\alpha\beta}$ when $|\mathbf{d}_{\alpha\beta}| < d^{excluded}$ and is equal to zero otherwise, where $d^{excluded} = 7\ nm$ and $k^{excluded} = 1690\ pN/\mu m$ [27]. This force is distributed to each of the four beads at the ends of the two line segments in proportion to their distance from the closest point, along $\mathbf{d}_{\alpha\beta}$. The excluded volume force is the main force which limits the timestep in the simulations to the value of $10^{-5}$ s in order to reduce unphysical bond crossings. We ensure that the number of bond crossings were sufficiently small to not influence our results, by measuring the number of bond crossings using the method of [96] and ensuring that this value does not significantly differ when using an order of magnitude smaller timestep.

Confining boundary forces in the $z$-direction act on any bead that is at $z < 0$ or $z > 0.2\ \mu m$ with a force of 1 $pN$ towards the interior of the simulation box [91].

## Membrane and motor pulling forces

Actin polymerization against the membrane at the cell leading edge, which is assumed stationary in this work, results in a pushing force (shown as $\mathbf{F}^{leading}$ in Fig 1A), that contributes to retrograde flow of the network. Focusing on the whole actin network dynamics, here we adopt a simple approach that results in the generation of a boundary pushing force on the actin network. The leading edge membrane force acts on all beads of each filament which has at least one bead in the top 0.5 $\mu m$ of the simulation box (region between -0.5 $\mu m$ and 0 in Fig 1A). Unless otherwise indicated, the total force per 1-$\mu m$ long filament was 1.5 $pN$ (0.136 pN per filament bead), on the order of force production by actin polymerization [50, 97] and close to the estimate of 1.7 pN filament stall force in keratocytes [60]. The pushing force is along the barbed to pointed end direction, as defined by the filament segment closest to the pointed end. Distributing the force equally to every bead in the filament avoids artificial filament bending and buckling at the leading edge, that we assume is not occurring under our conditions.

We implemented an approximate description of the molecular motor force leading to retrograde flow, similar to the pulling force by actin polymerization. We considered a motor force (shown as $\mathbf{F}^{motor}$ in Fig 1A) that acts in one of two modes: uniform or back, which are meant to represent the different spatial distributions of the retrograde flow motors (Fig 1A). In the uniform pulling mode, a force $\mathbf{F}^{motor}$ of magnitude 0.002 $pN$ is added to each actin bead in the simulation box, with direction along the positive $y$-axis. In the back pulling mode a force of magnitude 0.004 $pN$ with direction along the positive $y$-axis is added to each bead further than 3.25 $\mu m$ from the leading edge.

## Thermal forces

Thermal forces are calculated as a Gaussian distributed random value defined by

$$\langle \mathbf{F}_i^{thermal}(t) \rangle = 0 \tag{4}$$

$$\langle \mathbf{F}_i^{thermal}(t)\mathbf{F}_i^{thermal}(t') \rangle_{\alpha\beta} = 2k_B T \zeta_b \delta(t - t')\hat{I}_{\alpha\beta}, \tag{5}$$

where $\hat{I}_{\alpha\beta}$ is the second-order unit tensor ($\alpha$ and $\beta$ label the $x$, $y$ or $z$ directions).

## Focal adhesion region

We assume a simple frictional interaction between the actin network and focal adhesions. The nascent focal adhesion region is a capsule-shaped region as viewed in the $xy$ plane, with a height of 0.1 $\mu m$ as measured from the $z = 0$ surface as indicated in Fig 1E and with a width of 0.25 $\mu m$ and a length of 1.25 $\mu m$). Actin beads within this region can either be bound or

**Table 1. Simulation reference parameter values.**

| Parameter | Description | Value |
|---|---|---|
| $\Delta t$ | Simulation timestep [s] | $10^{-5}$ |
| $k^{actin}$ | Actin segment spring constant [pN/$\mu$m] | 1000 |
| $l_0$ | Actin segment spring eq. dist. [$\mu$m] | 0.1 |
| $l_p$ | Actin filament persistence length [$\mu$m] | 17.0 |
| $k^{crosslink}$ | Crosslinker spring constant [pN/$\mu$m] | 100 |
| $l_0^{crosslink}$ | Crosslinker spring eq. dist. [$\mu$m] | 0.035 |
| $k^{excluded}$ | Excluded volume interaction constant [pN/$\mu$m] | 1690 |
| $d^{excluded}$ | Diameter of actin filament [nm] | 7 |
| $\zeta_b$ | Friction coefficient [pN s/$\mu$m] | 0.108 |
| $k_{FA,bind}$ | Focal adhesion binding rate constant [1/s] | 10.0 |
| $k_{FA,unbind}$ | Focal adhesion unbinding rate constant [1/s] | 1.0 |
| $C_{dynamic}$ | Dynamic crosslinker concentration [$\mu$M] | 1.0 |

unbound to the nascent focal adhesion. Bound beads have a viscosity, $\eta_{FA}$, greater than or equal to that of the surrounding fluid, $\eta_{fluid}$, and so the viscous forces on these beads are higher. We define $\kappa_{FA} = \eta_{FA}/\eta_{fluid}$. Conversion between the bound and unbound states is handled via a discrete time implementation of the Gillespie algorithm [98] with binding and unbinding rate constants per bead as shown in Table 1. These rate constant values allow a significant fraction of the actin network within the focal adhesion region to be engaged with the focal adhesion, but are not strong enough to collapse the actin network into the adhesion along the $z$ direction. Mature focal adhesions were modeled in a similar manner to nascent adhesions, with higher $\eta_{FA}$ and were placed further away from the leading edge.

## Network generation, crosslinking, and disassembly

Filaments are added near the leading edge of the simulation box in the indicated region in Fig 1A at a fixed rate of 23.4/$s$ to maintain a concentration of 800 $\mu M$ of actin monomers at a retrograde flow speed of 30 $nm/s$. Each added filament has a length of 1.0 $\mu m$ and is comprised of 11 beads. The pointed ends of these filaments are initially at a position of $y = 0.5$ and have a random position in $x$ and $z$ that is within the bounds of the simulation box. The orientation of the filament in the $xy$ plane and with respect to the negative y-direction is uniformly distributed between -70˚ and 70˚. The angle of the filament in the $yz$ plane and with respect to the negative y-direction is uniformly distributed between -10˚ and 10˚, consistent with filament orientations in electron microscopy tomograms [15].

Upon addition of a filament, up to five permanent crosslinker bonds are added between the newly added filament and neighboring filaments. These connections simulate the connectivity through crosslinkers such as the Arp 2/3 complex. These bonds are only added to actin beads separated by a distance between 0.03–0.04 $\mu m$ (i.e. around $l_0^{crosslink}$). Unless otherwise stated, these bonds are permanent and are only removed when one of the two actin beads to which this bond connects are removed from the simulation. The concentration of Arp2/3 complex in XTC cell lamellipodia was estimated to be 2.3 $\mu M$ [67], which corresponds to approximately one Arp2/3 complex per 0.9 $\mu m$ of an actin filament, close to the branch distance of 0.75–0.8 per $\mu m$ observed by electron microscopy of fibroblast lamellipodia [15]. Average actin filament length in the branch network within a $\mu m$ to the leading edge can be as small as 0.2 $\mu m$ [15, 99]. In a typical simulation with $\kappa_{FA} = 100$, permanent cross-links have an average density of one per 0.3 $\mu m$, a value comparable to these experimental measurements.

Dynamic bonds, meant to represent dynamic actin crosslinkers, such as $\alpha$-actinin or filamin, are continuously introduced and removed as the simulation progresses. New dynamic bonds are selected randomly from each pair of beads separated by a distance of 0.03–0.04 $\mu m$ not currently bonded with each other. We assume a fixed maximum number of dynamic crosslinks, $N_{dynamic}^{total}$, which we place on the network as long as such connections are allowed. Instead of reporting values of $N_{dynamic}^{total}$, we use the equivalent concentration, $C_{dynamic}$, over a volume of the simulation box 9.5 $\mu m$ from the leading edge. In our simulations, we vary $C_{dynamic}$ between 1 and 30 $\mu M$, a range containing an estimate of the net $\alpha$-actinin and filamin crosslinker concentration of 2.9 $\mu M$, determined by using 100 $\mu M$ actin concentration for the entire cell and the experimentally determined $\alpha$-actinin and filamin crosslinker to actin molar ratios in macrophages [100]. Since the search for beads that are close to each other is computationally costly, the addition of dynamic crosslinkers is only performed every 0.025 $s$ instead of at every timestep. Each time dynamic crosslinkers are added, up to $N_{dynamic}^{total} - N_{dynamic}(t)$ new dynamic bonds are introduced randomly among allowed filament bead pairs, where $N_{dynamic}(t)$ is the number of dynamic bonds in the simulation at time $t$ before the addition of new dynamic crosslinkers. While this mechanism does not strictly satisfy detailed balance as in [101], we expect it to be sufficient (i.e. without artifacts) for the purposes of this work.

The disassembly of actin as well as crosslinker bonds is handled by selecting a bond lifetime, $\tau$, at the time $t$ of its creation, and removing it when the simulation advances beyond $t + \tau$. For actin segments, we assumed an age-dependent disassembly with a minimum lifetime of $\tau_{age} = 125$ $s$ followed by exponential decay of rate $r_{age} = 0.2/s$. Each filament segment thus has a lifetime calculated as $\tau = \tau_{age} - \ln(u)/r_{age}$ where $u$ is a random variable uniformly distributed between 0 and 1. If the removal of an actin segment results in a single-actin-bead filament, then this bead and any associated crosslinkers are also removed from the simulation. For dynamic crosslinkers, we assume an exponential lifetime distribution with decay rate $k_{dynamic}^{off} = 1/s$.

## Membrane pulling force by retrograde flow motors

A pulling force from the membrane was introduced in Fig 8 to represent the effect of pulling by membrane-bound motors such as myosin I [53, 54]. Only beads that are within the bottom $1/10^{th}$ of the simulation box ($z$ component of bead between $z = 0$ to 0.02 $\mu m$) and outside the focal adhesion region are pulled. The magnitude of the pulling force per bead was 0.02 $pN$, 10 times that of the original uniform pulling force. The pulling force was directed along the barbed-to-pointed end axis of the filaments instead of along the direction of retrograde flow.

## Branching simulation with angular restraint

In the branching simulations for Fig 8, new filaments are added as daughter filaments of length 1 $\mu m$ to actin beads of mother filaments (S4 Fig). In the non-branching simulations where filaments are introduced with their pointed ends at $y = 0.5$ $\mu m$, permanent crosslinkers formed primarily in the region between $0 - 0.5$ $\mu m$. To obtain a comparable simulation with branched filaments, the branch points were located on beads of mother filaments between $y = 0.25$ and 0.20 $\mu m$ (S4 Fig). The pushing force, $\mathbf{F}^{leading}$, is reduced to 30% of its value in other simulations to achieve similar retrograde flow speeds. Filaments were added at a 70˚ angle compared to the mother filament, at the same rate as in the simulations without branching. Added filaments were oriented along the lamellipodial plane (but overall covered the lamellipodial confining region along the $z$ axis). The angle between the mother and daughter filament, $\theta_{ijk}$, is restricted

to 70˚ by an angular force of the form

$$\mathbf{F}_m^{angle} = 2\epsilon_{angle}(cos\theta_{ijk} - cos\theta_0)\frac{\partial(\hat{d}_{kj} \cdot \hat{d}_{ji})}{\partial \mathbf{r}_m}, \quad m \in \{i, j, k\} \tag{6}$$

where $\epsilon_{angle} = 1\ pN\ \mu m$, $\theta_0 = 70˚$, $\{i, j, k\}$ are the indices of the actin beads which compose the branch, and $\hat{d}_{ji}$ is the unit vector from bead $i$ to bead $j$. We found that this value of $\epsilon_{angle}$ kept the branch angle between 60–80˚. Torsional rotation of daughter filaments around their mother filament is not constrained by this angular potential, but torsional rotation is restricted by the thin dimensions of our simulation box perpendicular to the branching plane as well as excluded volume interactions among filaments. This restriction is suggested from snapshots of branching simulations in Fig 8C where the majority of branches are in the $xy$ plane even far from their introduction near the leading edge.

## Mechanical characterization of simulated lamellipodial network

We measured the elastic properties of the actin network used in our model to compare to experimental values and to check the effect of softening the spring constant of actin bonds compared to the stiffness of actin filaments measured in experiments [27, 72]. The elastic properties were measured for a 4x4x0.2 $\mu m$ (WxHxD) patch of actin network. This patch was generated from a nascent focal adhesion simulation run in the pull uniform mode, with width 2 $\mu m$, using the reference parameter values of Table 1 with $\kappa_{FA} = 1.0$, evolved to steady state. The network from this focal adhesion simulation was subsequently equilibrated for 50 $s$ without pushing/pulling forces, filament addition/removal or crosslink changes (i.e. keeping all permanent and dynamic crosslinks that had formed). To form a 4 $\mu m$ wide actin network, we added a periodic image in the $x$-direction to the equilibrated actin network and reconnected periodic bonds across the newly generated interface. As a final step we cut the actin network to a 4x4x0.2 shape by removing actin beads outside the region defined by the boundaries at $y = -0.5\ \mu m$, $y = 3.5\ \mu m$ and at $x = -1.0\ \mu m$, $x = 3.0\ \mu m$, with bonds to beads outside this region also being removed.

To measure the uniaxial elastic modulus under extension, we moved actin beads that are initially within 0.25 $\mu m$ of either $y = -0.5\ \mu m$ or $y = 3.5\ \mu m$ at a fixed velocity in opposite directions along $y$ to give a strain rate of 0.2 $\mu m/s$, up to a maximal extension strain of 0.5 (S1(A) Fig). In these simulations we did not include actin filament disassembly or filament addition but allowed the existing dynamic crosslinks (that were formed under $C_{dynamic} = 1\ \mu M$) to unbind and rebind within the network. We kept excluded volume interactions and thermal fluctuations. We calculated the tensile stress summed over the network (not including the beads initially within 0.25 $\mu m$ of either edge) from the virial stress tensor, $\sigma_{ij}$, as $\sigma_T = \sigma_{yy} - 0.5(\sigma_{xx} + \sigma_{zz})$ [102]. Starting from the same initial network, we simulated stress versus strain for several different actin bond spring constants, permanent crosslink spring constant, and dynamic crosslink lifetimes (S1(B) Fig). As expected, we observed a nonlinear stress-stiffening behavior. The elastic modulus value we report is the ratio of the tensile stress divided by the strain at a strain of 0.2 [103]. For the reference parameters in this paper (Table 1, curve with $k^{actin} = 1000\ pN/\mu m$ in S1(B) Fig) the elastic modulus is 1355 $Pa$. We find that the network stiffens with increasing actin bond spring constant, with increasing spring constant of permanent crosslinkers, or by not allowing existing dynamic crosslinkers to unbind (remaining on the order $kPa$ for the range of parameter changes in S1(B) Fig).

We also simulated actin networks under compression up to a compressive strain of 0.5 (S1(A) Fig). For these simulations, we report $-\sigma_T$ (the compressive stress) since the value of $\sigma_T$ is negative under compression. For the simulation parameters used in the majority of our

simulations we find the modulus is 663 $Pa$ under compression. The compression modulus is smaller than the modulus under extension (S1(C) Fig), reflecting the nonlinear elastic properties of actin filaments [104]. For compression simulations we also perform simulations without a $z = 0.2$ $\mu m$ boundary, which allows for the actin network to buckle (S1(D) Fig) as has been seen in live cells when membrane tension increases during the switch from rapid to slow protrusion [85, 86]. As the value of $C_{dynamic}$ increases, these networks become increasingly sheet-like (S1(D) Fig). The buckling of these networks without the stabilizing force of the boundary at $z = 0.2$ $\mu m$ is evident in the stress-strain plots in S1(E) Fig which have a peak at the onset of buckling.

The elastic modulus values of the simulated networks are not significantly far off from the value of 985 $Pa$ reported for Arp2/3 complex networks measured by AFM tip oscillations in vitro [104], or measurements of $1 - 10$ $kPa$ for the elastic modulus of actin tails of *Listeria monocytogenes* [105]. The simulated modulus value is significantly softer than the measured elastic modulus at the leading edge of crawling keratocytes, 34 $kPa$ [106], however we note that the latter measurement may have included the contribution of polymerization force and adhesions. Taking these numbers in consideration, and for reasons of computational efficiency, we thus proceeded with the reference parameter values of Table 1.

## Simulation

Brownian dynamic simulations were performed using a custom C++ code with OpenMP implementation for parallel calculation of forces. Using 12 cores on XSEDE/SDSC-Comet for 48 hours allows for $\sim 65$ $s$ simulation time depending on the density of the system. To reach steady state, simulations were typically restarted around five times resulting in a typical total simulation time of around 300 s.

## Supporting information

**S1 Fig. Quantification of the elastic properties of the actin network used in simulations.** (A) A sample of the actin network taken close to the leading edge from a simulation at $\kappa_{FA} = 1.0$ and uniform pulling is put under either extensile or compressive strain up to a value of 50% at a strain rate of 0.2 $\mu m/s$. (The diffusing filaments escaping the network are filaments that got disconnected when the network was cut.) Blue line segments represent permanent crosslinkers while green line segments represent dynamic crosslinkers. (B) Tensile stress $\sigma_T$ versus strain measured by extension of the network. The elastic modulus is calculated by dividing $\sigma_T$ at a strain of 0.2 by the strain. We investigate the effect of varying the actin bond spring constant $k^{actin}$, permanent crosslinker spring constant $k^{perm}$, and the dynamic crosslinker dissociation rate $k^{off}_{dynamic}$ on the elastic modulus. Other parameters used as in Table 1, with the blue curve having all parameters (including $k^{actin}$) as in Table 1. Slight prestress in the $k^{actin} = 5000$ $pN/\mu m$ curve can be seen as a non-zero value of $\sigma_T$ at zero strain. (C) Tensile stress $\sigma_T$ versus strain measured by compression of the network. Boundary confining forces at $z = 0$ and $z = 0.2$ $\mu m$ prevent the thin sheet from buckling along the $z$ direction. Elastic modulus values measured for compression are lower than the same conditions as in (A) and less dependent on the values of $k^{actin}$, $k^{perm}$. (D) Networks under compression without the stabilizing effect of a repulsive boundary at $z = 0.2$ $\mu m$ exhibit buckling. Increasing $C_{dynamic}$ results in more cohesive networks. Color indicates $z$ value. (E) Tensile stress $\sigma_T$ versus strain measured by compression of the network without a repulsive boundary at $z = 0.2$ $\mu m$. (TIF)

**S2 Fig. Actin filament tension and concentration in simulations leading to arcs and bundles.** (A) Images of actin network at different dynamic crosslinker concentration, same as in Fig 6B, with a combination of long-lived and dynamic crosslinkers and focal adhesion representing a mature adhesion. Uniform pulling was applied. Blue line segments represent permanent crosslinkers while green line segments represent dynamic crosslinkers. (B) Tension between actin filament beads (averaged over both time and the $z$-axis at steady state) for the corresponding simulations in A show that the actin filaments are mainly under compression. (C) Actin concentration values (averaged over both time and the $z$-axis at steady state) for corresponding simulations in A. The scale along the vertical ($y$) and horizontal ($x$) axes is in $\mu m$. (TIF)

**S3 Fig. Actin bundle arc formation at the lamellipodium boundary with pull back mode, for comparison to uniform pull mode in Fig 6.** (A) Snapshots (left) and concentration profiles (right) by a combination of permanent and dynamic crosslinkers. Increasing the number of dynamic crosslinkers results in tighter bundles of filaments while leaving the overall spatial concentration of filaments relatively unchanged and there is no bundling of filaments near the focal adhesion region. Blue (green) line segments represent permanent (dynamic) crosslinkers. (B) Same as panel A but replacing permanent crosslinkers with crosslinkers that have a finite lifetime of 20$s$. Bundling, together with a peak in actin concentration, occurs in front of the focal adhesion region at intermediate dynamic crosslinker concentrations. The resulting bundle is less compact compared to the uniform pull case (Fig 6B). In the simulation snapshots, the scale along the vertical ($y$) axis is in $\mu m$. Blue (green) line segments represent long-lived (dynamic) crosslinkers. All data calculated from simulations equilibrated for at least 165 s and averaged over at least 24 s. (TIF)

**S4 Fig. Branching simulation illustration.** In the branching simulations of Fig 8, branches are added to a randomly selected actin bead between $y = 0.25$ and $y = 0.20\ \mu m$, highlighted by the green region. The branch is held at a prescribed angle of $70°$ by an angular force, $\mathbf{F}^{angle}$, defined in Eq 6, which acts on the three beads comprising the angle. Pushing forces from the leading edge shown in blue act on all beads of the daughter filament. In this example, there are no pushing forces on the mother filament beads to which the daughter filament is connected because the mother filament does not have a bead extending into the leading edge region. (TIF)

**S1 Movie. Lamellipodium under pull uniform condition.** Steady state lamellipodium from Fig 2 under pull uniform conditions with $\kappa_{FA} = 50$ over 30 $s$ and other parameters as in Table 1. (MP4)

**S2 Movie. Cytochalasin D simulation.** Simulated effect of cytochalasin D addition over 100 sec. At time $t = 0$ in the movie, addition and pushing of filaments is stopped in a simulation with $\kappa_{FA} = 50$ in the pull back mode with parameters as in Table 1, that had previously reached steady state. Blue line segments represent permanent crosslinkers while green line segments represent dynamic crosslinkers. (MP4)

**S3 Movie. Varying dynamic crosslinker concentration in simulation with mature focal adhesion and permanent crosslinkers.** First row of Fig 6 for $C_{dynamic} = 1, 10, 30\ \mu M$, over 30 seconds. Blue line segments represent permanent crosslinkers while green line segments represent dynamic crosslinkers. (MP4)

**S4 Movie. Varying dynamic crosslinker concentration in simulation with mature focal adhesion and long-lived crosslinkers.** Bottom row of Fig 6 where long-lived crosslinker lifetime is 20$s$ and $C_{dynamic}$ = 1, 10, 30 $\mu M$, over 30 seconds. Blue line segments represent long-lived crosslinkers while green line segments represent dynamic crosslinkers.
(MP4)

**S5 Movie. Simulations showing formation of microspike-like bundles.** Microspikes formation as in Fig 7. Blue line segments represent permanent crosslinkers while green line segments represent dynamic crosslinkers.
(MP4)

## Acknowledgments

We thank Naoki Watanabe, Sawako Yamashiro, and Koseki Kazuma for discussions that inspired this work and Danielle Holz for discussions and feedback.

## Author Contributions

**Conceptualization:** David M. Rutkowski, Dimitrios Vavylonis.

**Data curation:** David M. Rutkowski.

**Funding acquisition:** Dimitrios Vavylonis.

**Investigation:** David M. Rutkowski, Dimitrios Vavylonis.

**Methodology:** David M. Rutkowski, Dimitrios Vavylonis.

**Project administration:** Dimitrios Vavylonis.

**Resources:** Dimitrios Vavylonis.

**Software:** David M. Rutkowski.

**Supervision:** Dimitrios Vavylonis.

**Validation:** David M. Rutkowski.

**Visualization:** David M. Rutkowski.

**Writing – original draft:** David M. Rutkowski, Dimitrios Vavylonis.

**Writing – review & editing:** David M. Rutkowski, Dimitrios Vavylonis.

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
