## [Decision Letter · Decision Letter 0]

14 Jun 2021

Dear Dr. Vavylonis,

Thank you very much for submitting your manuscript "Discrete mechanical model of lamellipodial actin network implements molecular clutch mechanism and generates arcs and microspikes" for consideration at PLOS Computational Biology.

As with all papers reviewed by the journal, your manuscript was reviewed by members of the editorial board and by several independent reviewers. In light of the reviews (below this email), we would like to invite the resubmission of a significantly-revised version that takes into account the reviewers' comments.

We cannot make any decision about publication until we have seen the revised manuscript and your response to the reviewers' comments. Your revised manuscript is also likely to be sent to reviewers for further evaluation.

Sincerely,

Qing Nie

Associate Editor

PLOS Computational Biology

Jason Haugh

Deputy Editor

PLOS Computational Biology

Reviewer's Responses to Questions

**Comments to the Authors:**

Reviewer #1: This manuscript introduces a well-defined model for simulating a stationary lamellipodium. The model is primarily based on Brownian dynamics and includes explicit representation of individual actin filaments and a nascent focal adhesion. This allows them to investigate local deformation and force development with a retrograde flow at steady state. Most of the results are consistent with previous experimental findings. However, there are still some parts that lack explanations. The comments are listed below.

< Major comments >

- It is assumed that permanent cross-linkers can represent Arp2/3. However, there is a structural difference between a branched formed by Arp2/3 and a cross-linking point between two filaments. In particular, it is expected that 70 deg formed by Arp2/3 is maintained by intrinsic bending stiffness. Torsional rotation would not be allowed. However, it seems that the cross-linkers in the model do not have such things. This may affect the mechanical and dynamic behaviors of the lamellipodium significantly. The authors should justify the use of a simple cross-linker for Arp2/3.

- Cross-linkers can bind only to actin beads, which means that distances between cross-linkers would be the multiple of 100 nm. This also means that the length of “branches” in this pseudo-branched network structure would be also the multiple of 100 nm. However, in real branched structures formed by Arp2/3, this is far from true. The authors should provide justification about the assumption of the binding between actin filaments and cross-linkers.

- The focal adhesion is assumed to be a non-dynamic structure without further maturation or a decrease in size. Then, it would just behave as a friction pad to the moving network rather than a thing that behaves as a mechanosensitive structure as well as a friction pad. This might have been a convenient, simple setup for the model, but it doesn’t account for the reality well. How would the dynamic focal adhesion change the results overall?

- In the last paragraph of the introduction (page 3), the authors should briefly provide some numbers or descriptions of the model and elaborate the quantitative definition of “wide, steady lamellipodia” as well as the speed of retrograde flow. Are these numbers dependent on cell types? What is the standard or criterion for their stability in terms of size, morphology, and lifetime?

- What is the boundary condition in z-direction? Further, the orientations of filaments in the xy plane are not normally distributed. As the authors mentioned, this simplifies the force generation by polymerization or membrane tether. However, as far as I understand, the model does not account for a plasma membrane component. How would that affect the average pushing force exerted on each filament (~15 pN) measured from experiments? Also, what is the reference for the 15 pN?

- In Figure 1, it would be better to illustrate the motor force and crosslink forces. Schematic representation of motors is missing too. It might be difficult to draw the leading force in the force balance equation, but a simple demonstration would still be helpful.

- What are the simple criteria to differentiate the motors that exert uniform force from those that generate back force? Are all beads farther than 3.25 μm from the leading edge subjected to back pulling forces? What is the relative magnitude of the two forces?

- The filament segments are removed by severing near the bottom of the network as they mature. The simulation is run for ~65 s as mentioned in the method section, but the severing is for aged filaments with a lifetime longer than 125 s. Is there a discrepancy?

- In Figure 2A, for both pull back and pull uniform cases, there is always a sharp increase in magnitude as well as fluctuations for the retrograde velocity far from the leading edge, what is the explanation for that? Is it due to a variation in density of actin filaments? It would make a sense in the case of highest focal adhesion strength where there is a reduction in density as shown in 2D, but the reference case does not have such a reduction in density.

- It is interesting that the authors did not find significant perturbation of local network density or velocity by the nascent focal adhesion. However, in vivo, the strong correlation between motility force and adhesion/density is very commonly observed during cell migration. (i.e., https://www.pnas.org/content/118/4/e2009959118). This might be due to the lack of stress fiber structure as well as maturation of focal adhesion. Further discussion might be helpful.

- Is it possible to add more points to make the curves in Figure 4 more continuous? This is also applicable to Figure 5A. Plus, would it be necessary to perform some averaging of the data in each condition (with multiple runs)?

- If the two modes of pulling force lead to such a minimal difference in terms of the development of viscous and external forces as well as retrograde flow speed, then is it better to reconsider the initial assumption of the two pulling modes? Also, the switch of tension to compression is the same in both cases. The effect is negligible. Are the results in Figure 6 independent of the pulling modes as only the uniform pulling is implemented?

- The authors showed the formation of actin arcs near the focal adhesion. However, the actin arcs have been observed at the interface between lamellipodia and lamella without large focal adhesions. It seems that dense myosin population at the interface induces such arc formation rather than focal adhesion. Nevertheless, the observation of arc formation in this study was highlighted even in the title. The authors should demonstrate that a relationship between arc formation and focal adhesion by citing enough experimental studies.

< Minor comments >

Page 3, Line 59: “many have simplifying assumptions and do not include discrete filament force…” -> “simplifying” should be “simplified”. This should also be corrected in the rest of the manuscript.

Page 3, Line 77: “Here we model of a piece of the lamellipodium actin network at the filament level” -> “model of” seems to be wrong grammatically.

Page 3, Line 92: “filopodia” -> spelling. (Also, Page 15, Line 376).

Page 13, Line 336: “with their axis along”, possibly better to use “with their axes along”.

Reviewer #2: This is a very interesting study by Rutkowski  and Vavylonis, where they used coarse-grained molecular dynamics simulations to investigate the emergence of different actin network organizations within a steady-state lamellipodium. The authors used the well-studied XTC cells to benchmark their simulations in order to study the relationship between retrograde flow speed, focal adhesion strength, and force generation in lamellipodial actin networks. One of the most important findings are the conditions under which microspikes, actin arcs and bundles emerge in actin networks. Especially the emergence of lamellar arcs and stress fibre like architectures are novel. Although the results are well presented with high quality figures and movies, there are shortcomings in the clarity of explanation and comparison with experiments. I therefore suggest the authors address the following points in a suitably revised manuscript.

- It is unclear how the authors deduce the molecular composition of the actin network from experimental observation. For instance how do the authors choose the density of actin, relative abundance of Arp2/3 and dynamic crosslinks? How are these choices grounded in experimental observation.

- Typically myosin-II minifilaments are excluded from lamellipodial region and are found in the lamellar region. Therefore the case of uniform pulling does not make biological sense. Could the authors justify this choice? I am not aware of any experimental data that show contractile motors in the lamellipodium. Its also unclear why the motor pulling forces are acting vertically downward. Shouldn’t they act parallel to the filaments which will lead to both vertical and horizontal components?

- Retrograde flow speed depend both on motors forces and actin polymerisation. How are their relative contributions modelled?

- In Fig. 2 it is not surprising that in the focal adhesion region actin flow is reduced. The result is somewhat obvious since viscosity is increased at the sites of adhesion.

- The authors find that the F-actin flow profile (Fig. 2) is non-monotonic. Could they compare this to existing experimental data? Various groups have measure F-actin flow profiles in lamellipodia.

- The authors say "While the network velocity and density are not significantly perturbed locally by the 218 presence of a nascent focal adhesion” - Fig 2 seems to suggest otherwise

- How are pushing and pulling forces computed in Fig 4? I thought these were input parameters to the model (as defined in Fig. 1)

- I found the results in Fig. 6 and 7 very interesting, especially how actin bundles emerge within a disordered network. The authors may want to discuss these results in light of recent experimental studies - Vignaud et al Nature Materials 20:410 (2021), Lehtimaki et al eLife 10:e60710(2021). How do the length of the bundles in Fig. 7 compare with the typical size of microspikes?

**Have the authors made all data and (if applicable) computational code underlying the findings in their manuscript fully available?**

Reviewer #1: Yes

Reviewer #2: **No: **The code is not deposited to a public repository.

PLOS authors have the option to publish the peer review history of their article (what does this mean?). If published, this will include your full peer review and any attached files.

Reviewer #1: No

Reviewer #2: No
---

## [Decision Letter · Decision Letter 1]

30 Sep 2021

Dear Dr. Vavylonis,

We are pleased to inform you that your manuscript 'Discrete mechanical model of lamellipodial actin network implements molecular clutch mechanism and generates arcs and microspikes' has been provisionally accepted for publication in PLOS Computational Biology.

Best regards,

Qing Nie

Associate Editor

PLOS Computational Biology

Jason Haugh

Deputy Editor

PLOS Computational Biology

Reviewer's Responses to Questions

**Comments to the Authors:**

Reviewer #1: The authors addressed my comments and questions well by running additional simulations or including additional figures. I recommend acceptance.

Reviewer #2: The authors have carefully considered my comments, and have satisfactorily addressed all of them. I recommend the paper for publication.

**Have the authors made all data and (if applicable) computational code underlying the findings in their manuscript fully available?**

Reviewer #1: Yes

Reviewer #2: Yes

PLOS authors have the option to publish the peer review history of their article (what does this mean?). If published, this will include your full peer review and any attached files.

Reviewer #1: No

Reviewer #2: **Yes: **Shiladitya Banerjee

---

## [Editor Report · Acceptance letter]

13 Oct 2021

PCOMPBIOL-D-21-00824R1

Discrete mechanical model of lamellipodial actin network implements molecular clutch mechanism and generates arcs and microspikes

Dear Dr Vavylonis,

I am pleased to inform you that your manuscript has been formally accepted for publication in PLOS Computational Biology. Your manuscript is now with our production department and you will be notified of the publication date in due course.

With kind regards,

Zita Barta
